# Pneumolysin boosts the neuroinflammatory response to *Streptococcus pneumoniae* through enhanced endocytosis

Sabrina Hupp [1]✉, Christina Förtsch[2], Franziska Graber[3], Timothy J. Mitchell [4] & Asparouh I. Iliev [1]✉

In pneumococcal meningitis, bacterial growth in the cerebrospinal fluid results in lysis, the release of toxic factors, and subsequent neuroinflammation. Exposure of primary murine glia to *Streptococcus pneumoniae* lysates leads to strong proinflammatory cytokine and chemokine production, blocked by inhibition of the intracellular innate receptor Nod1. Lysates enhance dynamin-dependent endocytosis, and dynamin inhibition reduces neuroinflammation, blocking ligand internalization. Here we identify the cholesterol-dependent cytolysin pneumolysin as a pro-endocytotic factor in lysates, its elimination reduces their proinflammatory effect. Only pore-competent pneumolysin enhances endocytosis in a dynamin-, phosphatidylinositol-3-kinase- and potassium-dependent manner. Endocytic enhancement is limited to toxin-exposed parts of the membrane, the effect is rapid and pneumolysin permanently alters membrane dynamics. In a murine model of pneumococcal meningitis, mice treated with chlorpromazine, a neuroleptic with a complementary endocytosis inhibitory effect show reduced neuroinflammation. Thus, the dynamin-dependent endocytosis emerges as a factor in pneumococcal neuroinflammation, and its enhancement by a cytolysin represents a proinflammatory control mechanism.

*Streptococcus pneumoniae* (pneumococcus) causes severe and deadly infections such as pneumonia, sepsis, and meningitis. Pneumococcus is a common asymptomatic colonizer of the human nasopharynx; 20–50% of children and 5–20% of adults are carriers of pneumococcus, and its potential to become invasive depends on the ability of the host immune system to control the load of colonizing bacteria[1]. A reactive immune system and balanced commensal microbiota effectively control or clear pneumococci before they become pathogenic[2]. As soon as an imbalance occurs, pneumococci become invasive and cross epithelial barriers to invade the bloodstream or directly enter sterile sites, such as the brain[3]. *S. pneumoniae* produces various virulence factors that contribute to its colonization, transmigration through barriers,

and protection against the immune system. The protein toxin pneumolysin (PLY) plays a key role during these steps of infection[4].

During the initial stage of pneumococcal meningitis, bacteria multiply rapidly in the cerebrospinal fluid[5] and benefit from the absence of resident immune surveillance[6]. Bacterial growth rapidly reaches the lytic phase, leading to the release of pathogenic factors such as PLY, capsular components, and bacterial cell wall fragments (peptidoglycans)[7]. These factors trigger a strong inflammatory response in the meninges and underlying cerebral cortex[8], where microglia and astrocytes that express pattern recognition receptors (PRRs) represent the first line of defense[9–11]. Outside of the brain, innate responses against *S. pneumoniae* involve Toll-like

[1]Institute of Anatomy, University of Bern, Bern, Switzerland. [2]Institute of Pharmacology, University of Würzburg, Würzburg, Germany. [3]Institute of Pathology, University of Bern, Bern, Switzerland. [4]School of Immunity and Infection, College of Medical and Dental Sciences, University of Birmingham, Edgbaston, Birmingham, UK. ✉e-mail: sabrina.hupp@ana.unibe.ch; asparouh.iliev@ana.unibe.ch

receptors (TLRs) and the nucleotide-binding oligomerization domain-like receptor Nod2 (member of the NLR family)[12–14].

PLY is a major pathogenic factor that impairs the clinical course of pneumococcal infections in the brain[4,15]. PLY is a protein toxin released from bacteria through autolysis that binds to cholesterol in membranes of the host, building circular prepores of 30-50 monomers and eventually generating lytic pores[16]. While few host tissue cells are lysed, transient pore formation and nonlytic changes initiate substantial changes in cells and tissues[17,18]. The toxin must be capable of forming intact pores to cause these cellular changes (nonpore-forming PLY mutants have no cellular effects), but how most cells avoid cellular lysis remains unclear. Possible mechanisms include membrane repair and the shedding of toxin-loaded microvesicles from the cell surface[19]. PLY also has proinflammatory effects[20], but these effects cannot be reduced to its interaction with one specific receptor, despite some existing evidence[21,22].

The uptake of extracellular material into the cell via endocytosis enables cellular communication with its surroundings, the removal of damaged membranes, and the presentation of foreign extracellular molecules to multiple intracellular PRRs. There are at least 10 different molecular mechanisms of cargo uptake. Of these mechanisms, phagocytosis/pinocytosis, clathrin-mediated endocytosis, and uptake into caveolae are among the most well-studied pathways. Both pathogens and bacterial toxins can hijack endocytosis to overcome local defenses and translocate through barriers. For example, the anthrax toxin enters cells via a raft dependent, clathrin-mediated mechanism; clostridial toxins access the cytosol via dynamin dependent, clathrin-independent endocytosis; and vaccinia viruses use macropinocytosis to promote their own uptake[23–25]. The foodborne intracellular pathogen *Listeria monocytogenes* utilizes its cholesterol-dependent cytolysin listeriolysin O (LLO) to enhance endocytosis and facilitate its entry into cells. Subsequently, *L. monocytogenes* uses LLO to escape from the endocytic vacuole and spread further[26].

Here, we demonstrate that in mixed primary glial cultures, *S. pneumoniae* lysates had proinflammatory effects exclusively via Nod1. This differs from other nonneural tissues, where the effect is Nod2 dependent. A dynamin-dependent endocytotic pathway was critical in this process, and PLY was the key bacterial trigger of this pathway. PLY hijacks the endocytotic machinery to enhance the neuroinflammatory response, which represents a mechanism utilized by bacteria in the host.

## Results

### Neuroinflammation is dependent on dynamin and Nod1
We incubated mixed glial cell cultures (a standard model system for the brain environment that surrounds neurons, containing both astrocytes and microglia) with pneumococcal lysates from wild-type D39 (serotype 2) pneumococci (D39 wt). The bacterial lysate amounts in all experiments were equivalent to $1 \times 10^7$ CFU/ml. Bacterial numbers in the cerebrospinal fluid of patients with pneumococcal meningitis range from $4.5 \times 10^3$ to $3 \times 10^8$ CFU/ml, normally above $1 \times 10^7$ CFU/ml[27,28]. Exposure of glial cells to pneumococcal lysates for 24 h yielded a strong increase in the major proinflammatory cytokines TNF-α and IL-6 (both have the strongest predictive value in systemic bacterial infections and correlate with worse outcomes[29]) and the polymorphonuclear chemokine CXCL-2 (chemokine (C–X–C motif) ligand 2, also called macrophage inflammatory protein 2 (MIP2)) (Fig. 1a). In contrast to findings in peripheral tissues, where *S. pneumoniae* acts on Nod2 receptors[30], Nodinitib-1 (a highly selective Nod1 inhibitor[31]) strongly inhibited cytokine release (Fig. 1a). GSK583 (a specific inhibitor of the Nod downstream effector RIP kinase 2 (receptor-interacting serine/threonine-protein kinase 2)) also blocked the inflammatory response. Nod1 and Nod2 are cytosolic receptors that rely on endocytosis for the delivery of their ligands into the cytosol (overview in Supplementary Fig. S1)[32].

Intracellular receptors require internalization of their ligands to elicit a response. Therefore, we analyzed the level of endocytosis in glial cells and investigated whether pneumococcal lysates enhance endocytosis or utilize only the standard endocytotic rate. For this purpose, we used the FM4-64 fluorescent dye, as this dye binds to the outer leaflet of the lipid bilayer and, upon internalization and acidification of membrane-derived vesicles, its fluorescence strongly increases[33]. The D39 wt lysate substantially enhanced endocytosis (Fig. 1b, curve normalized to mock-treated controls; Supplementary movie M1) in a dynamin-dependent manner. The effect was similar in both microglia and astrocytes (Supplementary Fig. S2). In a cytokine release assay, the bacterial ability to stimulate the production of the proinflammatory cytokines TNF-α and IL-6 was strongly blocked by the dynamin inhibitors Dyngo4a and Dynasore (Fig. 1c). The inhibition of the chemokine CXCL-2/MIP2 following dynamin inhibition was milder but still significant (Fig. 1c).

For all further analyses, we focused on TNF-α and IL-6 as the most clinically relevant predictive cytokines[29]. We confirmed that endocytosis inhibitors did not alter the viability of the cells (Supplementary Fig. S3).

We tested the abilities of several other specific inhibitors of innate inflammation, namely, inhibitors of TLR2, TLR4, and TLR9, to block the neuroinflammatory glial response to lysates, but we failed to observe any inhibitory effects, except for IL-6 secretion by hydroxychloroquine (known to modulate intracellular trafficking) (Supplementary Fig. S4).

### Pneumolysin is the pro-endocytic factor of *S. pneumoniae*
Next, we looked for the bacterial factor responsible for the enhanced endocytosis. We identified a well-known pathogenic factor—the cholesterol-dependent cytolysin pneumolysin—as the major pro-endocytotic factor in pneumococci. Pneumococci that did not express PLY (D39 ΔPLY) showed a substantially decreased ability to enhance endocytosis (Fig. 2a, b).

PLY (and some other CDCs, such as PFO and LLO[34,35]) is important as a proinflammatory factor in epithelial and glial cells[20,36], but the underlying mechanisms remain elusive. To verify that PLY acts similarly in our system, we exposed mixed glia to D39 wt and D39 ΔPLY lysates and observed a substantially reduced release of TNF-α and IL-6 when PLY was lacking (Fig. 2c). As a comparison, we used recombinant PLY in equivalent amounts to those in the lysates (comparison based on the permeabilization capacity of the lysates) and observed only a mild increase in IL-6 and none in TNF-α (Fig. 2d). Recombinant PLY from *E. coli* may contain minimal traces of LPS, which can slightly stimulate cytokine release (Supplementary Fig. S5). Therefore, we performed these and all follow-up experiments with recombinant PLY either after additional LPS removal or with the addition of polymyxin B, which inactivates LPS.

The time frame of inflammatory cytokine release demonstrated a linear increase when wild-type D39 lysates were used even at 24 h, while D39 ΔPLY lysates reached a cytokine stimulation plateau at 12 h after challenge (Supplementary Fig. S6).

### PLY boosts endocytosis in a cholesterol-, pore-dependent way
To verify the results of the knockout experiments, we examined the ability of recombinant PLY to enhance endocytosis. Sublytic concentrations of PLY (concentrations yielding <5% lysis, 2 HU/ml in our preparations)[37] enhanced endocytosis, and this effect was fully blocked by preincubation of the toxin with cholesterol (Fig. 3a), confirming that cholesterol binding was essential. Mixed glial cultures comprise both microglia and astrocytes, which can be discriminated excellently in live-imaging conditions[38]. Both cell types demonstrated increased endocytosis in response to PLY (Fig. 3b). PLY was also capable of enhancing endocytosis in nonbrain cells, such as HEK293 cells (Fig. 3c). The pore formation capacity of PLY was essential for these effects, as the application of either pore-deficient mutants (Δ6 and W433F PLY mutants[39])

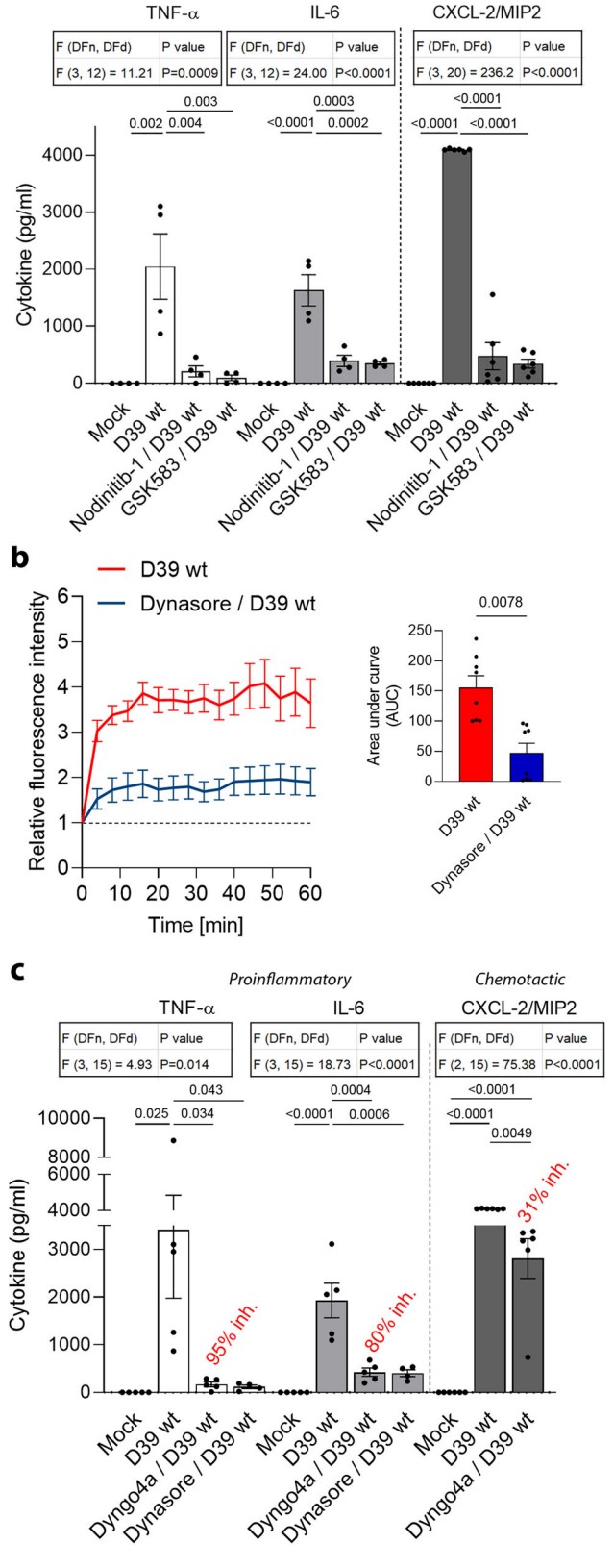

**Fig. 1 | Reduction of the neuroinflammatory response against pneumococcus by Nod1, RIP2 kinase, and dynamin inhibition. a** Release of TNF-α, IL-6 and the neutrophil-specific chemokine CXCL-2 (MIP2) from mouse mixed glial cultures 24 h after stimulation with the D39 wild-type lysate (D39 wt; $1 \times 10^7$ colony-forming units (CFU)/ml) is substantially reduced by Nodinitib-1 (2 μM; Nod1 inhibitor) and GSK583 (5 μM; RIPK2 inhibitor). **b** Elevated endocytosis (FM4-64 assay) in murine mixed glial cells after challenge with the D39 wild-type lysate (D39 wt) is inhibited by the dynamin inhibitor Dynasore (10 μM). Curves are normalized to the mock-treated controls. Statistically significant differences in the area under the curve are represented in the bar graph. **c** Elevated release of TNF-α, IL-6, and CXCL-2 (MIP2) after stimulation with the D39 wt lysate is inhibited by pretreatment of the cells with dynamin inhibitors (all 10 μM). All experimental groups were tested in parallel in each repeat. Differences in cytokine release in D39 wt-treated cultures between **a** and **c** were due to natural variations among cultures and bacterial lysates. All values represent the mean ± SEM, each dot symbol indicates an independent experiment (**a** TNF-α and IL-6 $n = 4$, CXCL-2/MIP2 $n = 6$; **b** $n = 8$; **c**. TNF-α and IL-6 $n = 5$, with exception of Dynasore/D39 wt $n = 4$, CXCL-2/MIP2 $n = 6$), exact $p$-values are indicated (if significant), one-way ANOVA with Tukey's post-test (**a, c**), Wilcoxon matched pairs test (**b**), all tests are two-tailed. Source data are provided as a Source Data file.

membrane shedding, changing the properties of the membrane. To prove the enhanced endocytosis directly, we incubated glia with PLY and visualized it via transmission electron microscopy (TEM). Both astrocytes (Fig. 4a) and microglia (Fig. 4b) demonstrated substantially increased amounts of smaller vesicular structures (<500 nm in size, corresponding to clathrin-coated vesicles, caveolar vesicles, and early endosomes[40]) in the cytosol (Fig. 4).

In time-lapse imaging experiments, we followed the membrane dynamics of microglia (Fig. 5a). In untreated cells, dynamic but balanced (in all directions) membrane waves were observed. Once challenged with sublytic amounts of PLY, these waves lost their chaotic character and became centripetal; this phenomenon was concurrent with shrinking of the cells and clustering of the cellular vesicles around the nucleus (Fig. 5a and Supplementary movie M2). FM 4-64-positive endosomal vesicles appeared along the membrane waves (Fig. 5a, white arrows). In electron microscopy, treated microglia demonstrated a highly elevated density of membrane waves packed with multiple vesicles (Fig. 5b, red arrows).

### PLY boosts endocytosis in a dynamin-, PI3K-, and K⁺-dependent way

We next studied the effect on endocytosis enhancement of various membrane-associated endocytosis-relevant mechanisms, also relevant to known/potential toxin effects (Fig. 6; all graphs show curves normalized to mock- or inhibitor controls, all nonnormalized raw data are presented in Supplementary Fig. S7 and in the repository file/source data file). In Table 1, we listed all tested inhibitors/inhibitory conditions with references to their effects, used and relevant concentrations, and possible link to PLY effects. Briefly, we tested dynamin inhibitors (Dynasore, MiTMAB, and Dyngo4a) with different chemical properties, affinities and specificities for dynamin[41]), pinocytosis/submembranous acidification zone formation inhibition (amiloride[42]), and extracellular ion depletion (potassium and calcium; PLY causes many calcium-dependent cellular effects[43]. Extracellular calcium depletion leads to an enhanced lytic capacity of PLY[44], and we adapted our assays in calcium-free buffers to maintain equivalent cytotoxicity in all treatment groups.), PIP/PIP2/PIP3 turnover enzyme inhibitors (PLC, PI3K and PTEN), endocytosis- and toxin-relevant kinases inhibitors (Src, Akt, PKC) and ouabain—an inhibitor of the Na/K-ATPase (Fig. 6). Similar to the experiments with the D39 lysates, inhibition of dynamin GTPases diminished the changes in endocytosis (Fig. 6) in both astrocytes and microglia (Supplementary Fig. S8; discrimination by morphology[38]). There was no difference in the acute PI permeabilization by PLY in the presence versus absence of Dynasore (Supplementary Fig. S9). Depletion of potassium diminished endocytosis, as did the PI3K

or a domain 4 form of the toxin (the membrane-binding nonpore-forming domain of PLY) failed to enhance endocytosis (Fig. 3d).

### PLY acts rapidly and changes permanently membrane dynamics

The FM assay is a well-established method for evaluating endocytosis. Nevertheless, molecules such as PLY produce pores and initiate

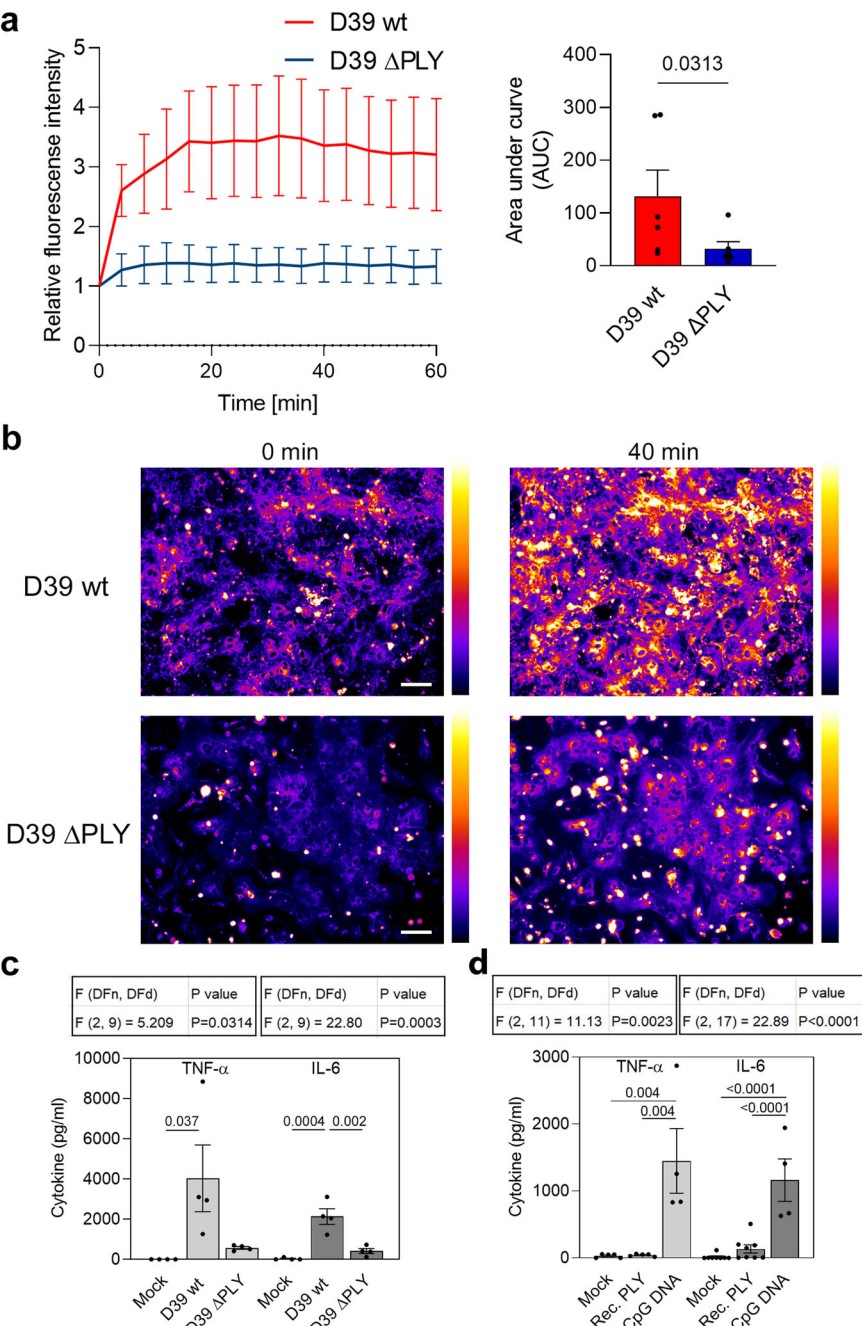

**Fig. 2 | Pneumolysin (PLY) is the pro-endocytic and proinflammatory factor in *Streptococcus pneumoniae* lysates but is not proinflammatory alone. a** The increase in endocytosis (FM 4-64 assay) was nearly completely abolished after exposure to lysates of D39 pneumococci lacking PLY (D39 ΔPLY). Curves are normalized to the mock-treated controls. The bar graph represents the area under the curve comparison. **b** Representative image of FM 4-64 fluorescence staining. In these images, the brighter stained cells represent microglia (saturated cells are masked) (pseudocolor Fire LUT (ImageJ)). Scale bar: 100 μm. **c** The neuroinflammatory response (TNF-α and IL-6) of murine mixed glial cultures 24 h after D39 lysate challenge is substantially diminished in absence of PLY (D39 ΔPLY). **d** LPS-free PLY at a dose of 2 HU/ml is unable to stimulate TNF-α release and only minimally stimulates IL-6 release. CpG DNA represents a positive control. All values represent the mean ± SEM, each dot symbol indicates an independent experiment (**a** $n = 6$; **c** $n = 4$; **d** IL-6 $n = 8$ (except CpG DNA $n = 4$), TNF-α $n = 5$ (except CpG DNA $n = 4$)), exact $p$-values are indicated (if significant). Wilcoxon matched pairs test (**a**), one-way ANOVA with Tukey's post-test (**c**), all tests are two-tailed. Source data are provided as a Source Data file.

inhibitor wortmannin, which blocks the phosphorylation of PIP2 to PIP3 (Fig. 6). The inhibition of another enzyme involved in PIP2 degradation, PLC, elevated PLY-dependent endocytosis, but surprisingly, the inhibitor of PTEN (leading to dephosphorylation of PIP3 to PIP2) failed to influence the process. The endocytosis stimulated by PLY was also enhanced by the blocker of the Na/K ATPase ouabain (Fig. 6). The activity of the inhibitors was validated in Supplementary Fig. S10.

## Endocytic boost by PLY is rapid and compartmentalized

To fit the time frame of endocytosis enhancement to the other known cellular effects of PLY—cell swelling[18], membrane depolarization (DiBac$_4$(3))[17], calcium influx, and extracellular vesicle shedding—we compared the exact occurrence of endocytosis enhancement (Fig. 7a). This was a rapid phenomenon, starting simultaneously or shortly after the elevated vesicle shedding and membrane depolarization and coinciding with calcium influx but preceding cell swelling (Fig. 7a). The

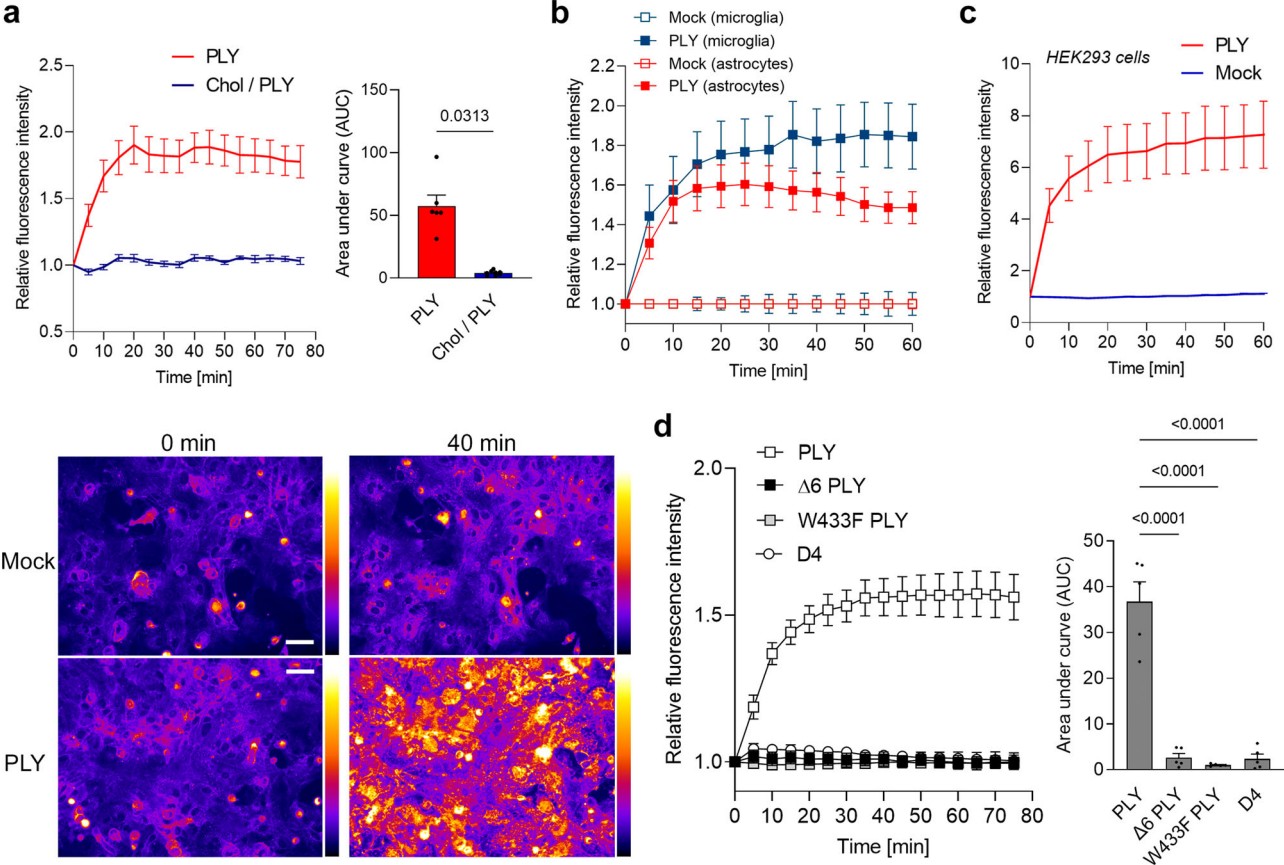

**Fig. 3 | Recombinant PLY enhances endocytosis in a pore-dependent manner.**
**a** Recombinant PLY (2 HU/ml) enhances endocytosis (FM 4-64 assay), and the process is blocked by preincubation of the toxin with cholesterol. Curves are normalized to the mock-treated cells. Representative images demonstrate the fluorescent intensity at corresponding time points (pseudocolor Fire LUT (ImageJ)). Scale bar: 50 μm. **b** Fluorescence intensity changes after PLY challenge (2 HU/ml) in microglia and astrocytes. All curves were normalized to the mock-treated cells. **c** Enhanced endocytosis by PLY in nonbrain cells such as HEK293 fibroblasts.

**d** PLY-mediated enhanced endocytosis is eliminated due to the loss of the pore-forming capacity of PLY in the Δ6 mutant (Δ6 PLY) and W433F mutant (W433F PLY). Similarly, the membrane-binding domain 4 of PLY (D4) does not alter endocytosis. All values represent the mean ± SEM, each dot symbol indicates an independent experiment (**a** n = 8 (Chol/PLY), n = 6 (PLY); **b** n = 11 (mock) and n = 13 (PLY); **c** n = 8; **d** n = 5), exact p-values are indicated (if significant), Wilcoxon matched pairs test (**a**), one-way ANOVA with Tukey post-test (**d**), all tests are two-tailed. Source data are provided as a Source Data file.

next question was whether the enhanced endocytosis was a generalized cell response because of the toxin/membrane interaction or a localized effect limited to the area of direct interaction. To answer this question, we established a glial culture on top of a porous membrane insert (pore size 1 μm) (Fig. 7b). The underside of the glia communicated only with the lower chamber, which was exposed to PLY. An enhanced FM 4-64 signal was observed at these porous openings, where the cells encountered the toxin (Fig. 7c, red curve), but the effect did not proliferate to the upper membrane of the cells, which communicated only with the toxin-devoid upper chamber (Fig. 7c, green curve). Once the toxin was added to the upper chamber, a total cellular endocytotic response was observed (Fig. 7c). This experiment suggests that enhanced endocytosis is an effect limited to the direct interaction point of the toxin with the membrane rather than a propagating response of the whole cell.

## PLY-GFP is internalized too, but dynamin-independently

Although PLY-enhanced endocytosis, there is evidence that the toxin itself can be removed by membrane shedding[19]. Using a functional GFP-tagged form of PLY, we analyzed whether PLY was internalized and whether the increased dynamin-dependent endocytosis affects its own internalization[45]. The toxin was indeed internalized, as determined by a 3D reconstruction of RFP-transfected astrocytes (Supplementary Fig. S11a and Supplementary movie M3). This process was, however, dynamin-independent (Supplementary Fig. S11b).

PLY did not colocalize with small (<1 μm), fluorescent, transferrin-positive vesicles (transferrin is a marker of classic receptor-mediated endocytosis) (Fig. 8). Colocalization was observed only in larger, heterogeneous structures, corresponding in size and structure to multivesicular bodies (Fig. 8, arrows). PLY did not colocalize with fluid-phase endosomes or late endophagosomes (using fluorescent dextran as a probe), caveolae-derived vesicles (using fluorescent albumin as a probe), or Arf6-immunopositive endosomes (Fig. 8). However, PLY moderately colocalized with the small flotillin-immunopositive vesicles (Pearson correlation coefficient r = 0.35, Fig. 8). These experiments confirmed that despite enhancing dynamin-dependent endocytosis, PLY itself was internalized in a dynamin-independent manner into a different compartment and thus could not contribute to the release of ligands from dynamin-dependent endosomes via pore formation.

## Endocytosis block reduces neuroinflammation in pneumococcal meningitis mice

Finally, to address the ability of endocytosis inhibition to modulate neuroinflammatory responses against *S. pneumoniae* in animals, we used the neuroleptic drug chlorpromazine, an endocytosis inhibitor with well-known permeability through the BBB[46]. We initially tested its effect against PLY-induced endocytosis. It was effective at low concentrations of PLY (1 HU/ml), while at higher concentrations of the toxin, CPZ surprisingly enhanced the cytotoxic effect of PLY and

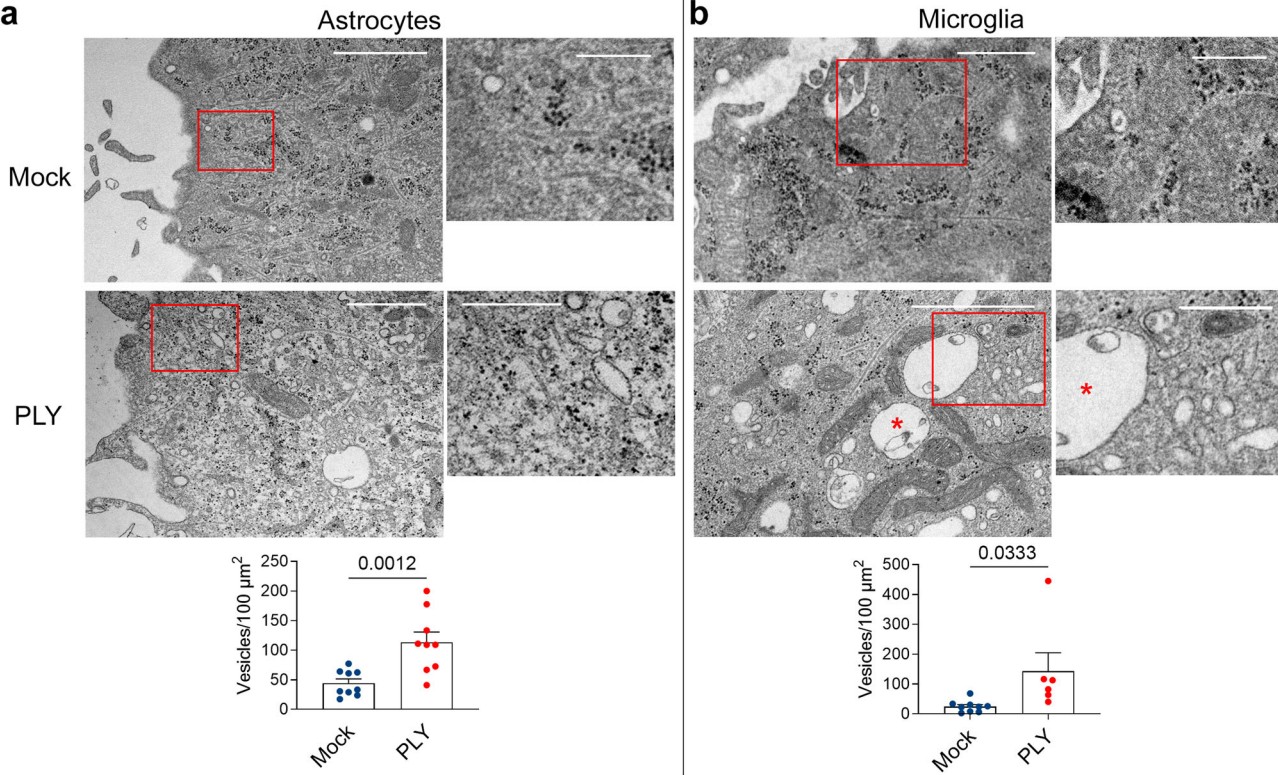

**Fig. 4 | Transmission electron microscopy of glial cells after recombinant PLY exposure.** TEM in astrocytes (**a**) and microglia (**b**) demonstrates elevated numbers of vesicular structures with sizes <500 nm, corresponding to clathrin-coated vesicles, caveolar vesicles, and early endosomes. The red square is magnified twofold in the smaller adjacent image. All values represent the mean ± SEM, each dot symbol indicates a single cell (astrocytes mock $n = 9$ cells, PLY $n = 6$ cells; microglia mock and PLY $n = 9$ cells), Mann–Whitney $U$-test, all test two-tailed, exact $p$-values are indicated (if significant). The scale bar in the lower magnification images is 1.5 µm, and the scale bar in the higher magnification images is 0.5 µm. Source data are provided as a Source Data file.

failed to block endocytosis (Supplementary Fig. S12). In tissues, cell lysis is rarely observed, and we thought that CPZ would lend at least partial support to our concept. Eighteen hours after the initiation of meningitis, the level of TNF-α was significantly inhibited by CPZ (the cytokine that was initially shown to be more strongly dependent on endocytosis (see Fig. 1d)), but the level of IL-6 was not (Fig. 9a), although IL-6 levels showed a decreasing trend. The inhibition was not as strong as that produced by dynamin inhibitors in culture. The bacterial counts in the brain, spleen, and blood remained similar between all groups after treatment with CPZ (Fig. 9b) and were not due to the antibacterial properties of CPZ described in earlier reports[47].

## Discussion

Here, we demonstrate for the first time that the proinflammatory effect of the cholesterol-dependent cytolysin pneumolysin in the brain is substantially mediated by its effect on cell trafficking, namely, by enhancing endocytosis. Furthermore, we identified the glial Nod1 receptor as a key neuroinflammatory PRR for lysed components of pneumococci.

The immune system responses in the brain differ from those throughout the body. The immune-privileged environment, the abundance of microglial cells, and the nearly complete lack of lymphatic structures increase the overall effects of innate immunity. Sustained neuroinflammation damages the brain, specifically neurons, in many neurological conditions, including Alzheimer's disease, multiple sclerosis, and brain trauma, among others[48]. In bacterial meningitis (pneumococcal and nonpneumococcal), inhibition of neuroinflammation improves the outcome[49]. Nevertheless, neuroinflammation in meningitis can also be beneficial—the inactivation of TNF-α and IL-6 signaling in mice leads to increased lethality and

neural deficits in surviving animals[50,51]. Neuroinflammation is thus important for eliminating pathogens, but if the response is too strong, it can also be harmful.

RIP kinase 2 inhibition in meningitis modulates the outcome of the disease[30], emphasizing the importance of the Nod/RIPK axis. Outside the brain, the PRR Nod2 plays a critical role, mediating the inflammation stimulated by pneumococci[52]. Our finding that Nod1 is a factor of neuroinflammation highlights a difference between neural and nonneural tissues. The ligands activating Nod1 and Nod2 are distinct, and it is assumed that Nod1 is predominantly activated by Gram-negative microorganisms, while Nod2 is predominantly activated by Gram-positive microorganisms[53]. Nod1 is known to bind γ-ᴅ-glutamyl-mesodiaminopimelic acid (iE-DAP) from Gram-negative bacteria but with a low affinity (Kd = 30 µM)[54]. Evidence from infant patients with Gram-positive sepsis, however, confirms that Nod1-receptor variants are relevant to the disease course[55]. Nod-like receptors sense not only peptidoglycan ligands but also other cellular changes, such as small GTPase activation and ER stress[52]. Taken together, these data indicate either the existence of a novel Nod1 ligand, the occurrence of complex cellular changes leading to receptor activation or undescribed off-target effects of the Nod1 inhibitor.

Several other innate immune PRRs, such as TLR2, TLR4 and TLR9, have been implicated in the pathogenesis of bacterial, and specifically pneumococcal, brain disease, but in most cases in isolated cell culture conditions. Bacterial DNA can be neurotoxic by activating Toll-like receptor 9 (TLR9)[56], although the effect is only apparent after at least 48 h. TLR2-knockout animals suffer from weaker neuroinflammation early in meningitis, followed by a stronger response later[57]. Isolated peritoneal macrophages from TLR2-knockout animals, however, fail to demonstrate any change in their inflammatory response toward pneumococci, suggesting more complex crosstalk at the cellular and

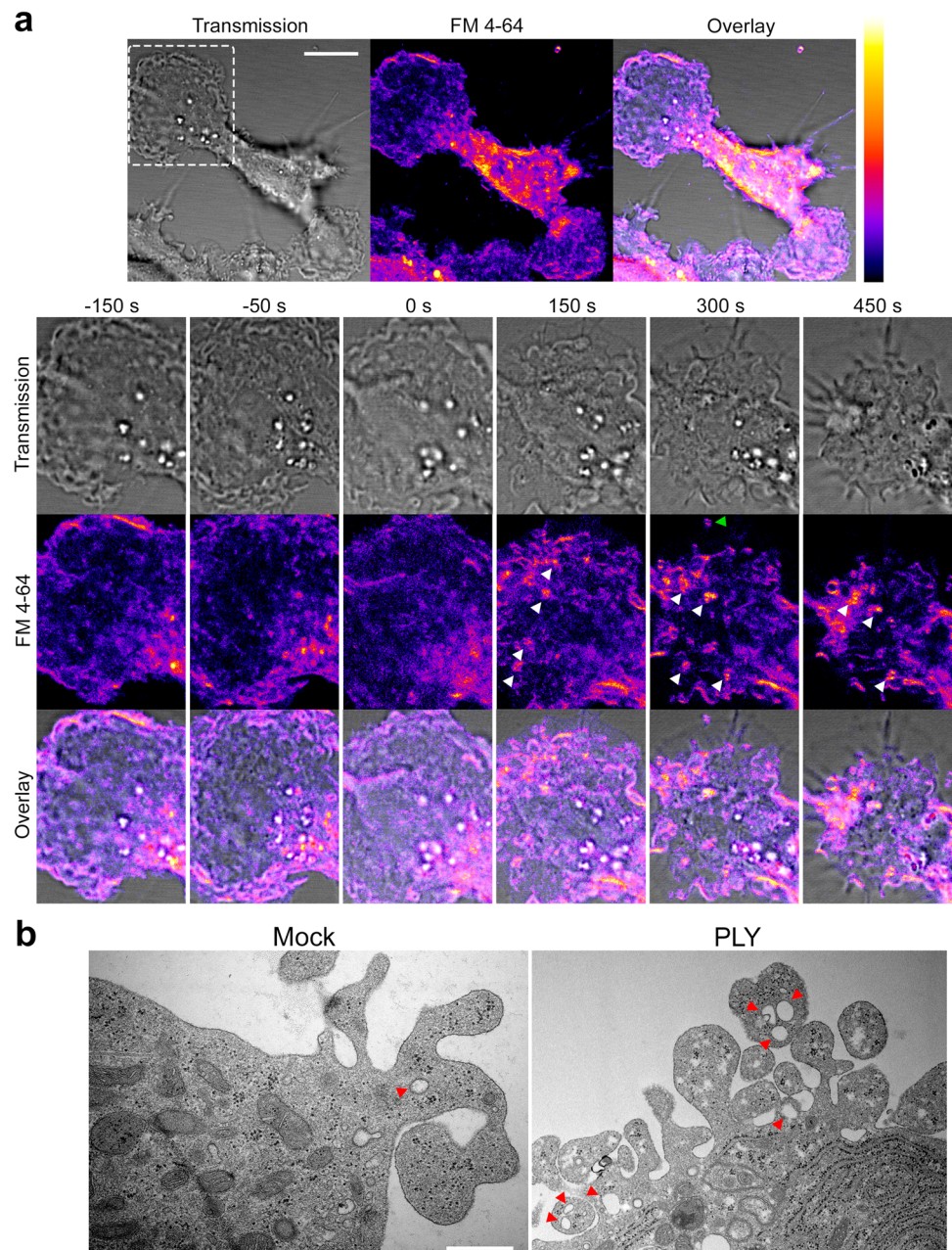

**Fig. 5 | Time-lapse imaging of microglial membrane alterations after recombinant PLY exposure. a** High-resolution video microscopy of microglia in transmission and fluorescent imaging (FM 4-64). The fragment in the first image is magnified in the following series (0 s−time-point of treatment; see also Supplementary movie M2). Before treatment, the membrane demonstrates balanced random waves. Following challenge with 2 HU/ml PLY, the waves redirected toward the body of the cell and slowed down, followed by retraction and clustering of the vesicular structures in the center of the cell. White arrows indicate newly occurring endosomal vesicles, and the green arrow indicates released membrane vesicles. All newly occurring FM-positive endosomal structures are observed adjacent to membrane waves (pseudocolor Fire LUT (ImageJ)). Scale bar: 10 μm. **b** Electron microscopy of the microglial surface demonstrates a massive increase in membrane protrusions, with a strong increase in the number of vesicular structures (red arrows). Scale bar: 500 nm.

organismal levels[58]. Therefore, we are careful in interpreting our cell culture findings that TLR2 and TLR4 inhibition does not block pneumococcal neuroinflammation in culture. Several works have suggested the involvement of TLR2 and TLR4 signaling in PLY recognition, but the direct interaction of PLY with these receptors has not been confirmed thus far[21,59]. Our experiments with polymyxin B and LPS removal kits suggest that these results may be due to contamination with minimal amounts of LPS in the recombinant toxin preparations. Another possibility is that PLY, similar to Nod1, can influence the intracellular trafficking of various TLR ligands. We do not exclude the possibility that other proinflammatory effects of PLY, such as

inflammasome activation[60], actively modulate inflammation together with endocytosis modulation.

The extensive analysis of various inhibitors relevant to different pro-endocytic pathways (see Table 1) revealed three effective mechanisms−dynamin inhibition, potassium depletion and phosphatidylinositol-3-kinase (PI3K) inhibition. Potassium depletion leads to membrane hyperpolarization, which can antagonize membrane depolarization by PLY[17]. Membrane depolarization, however, can only initiate endocytosis in excitable neural tissues in a $Ca^{2+}$-dependent manner[61], which is not the case in our paradigm. Extracellular potassium depletion can also slowly deplete intracellular

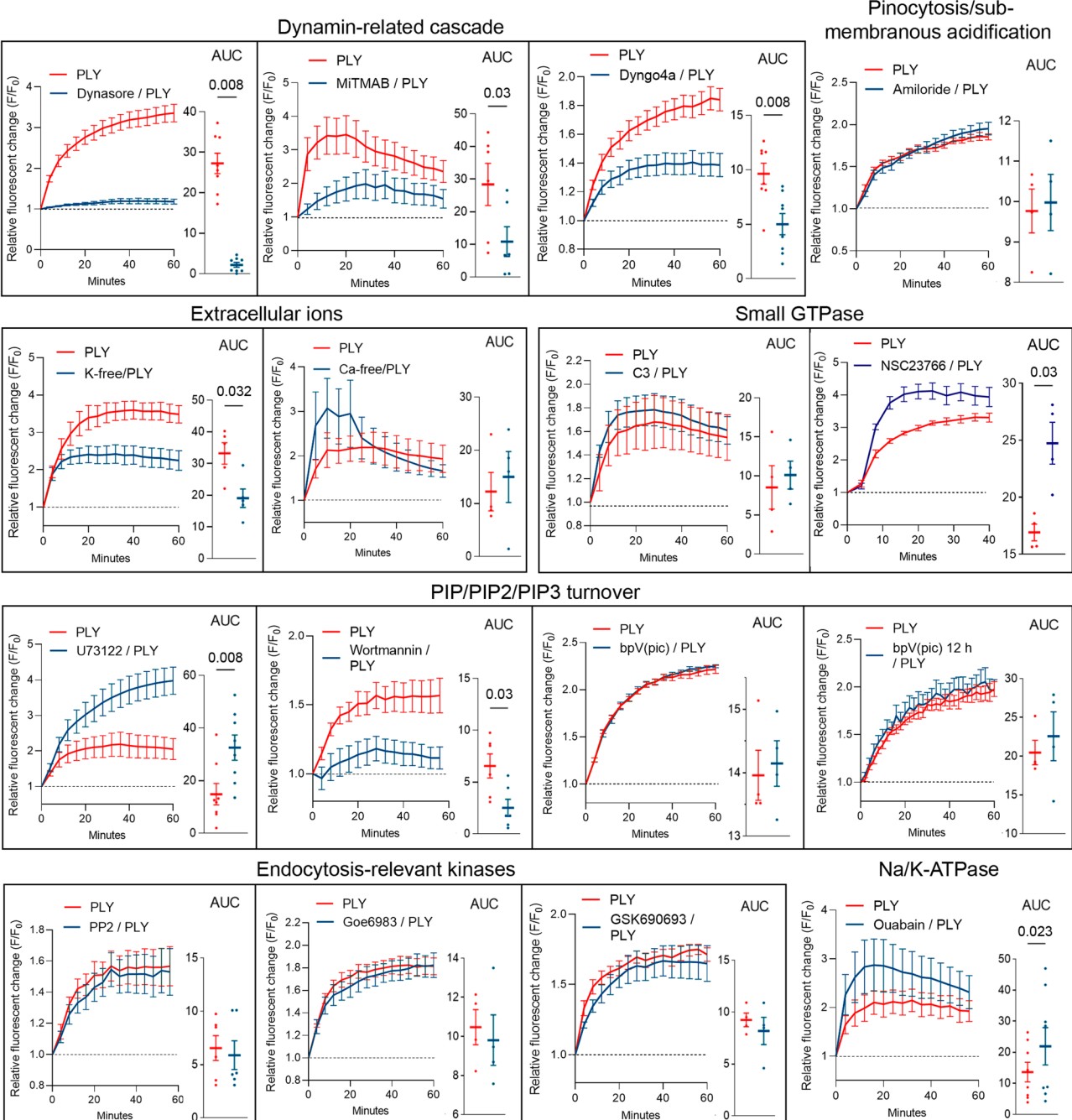

**Fig. 6 | Molecular mechanism of enhanced endocytosis by recombinant PLY.**
Curves normalized first to timepoint 0, followed by normalization to mock (vehicle)-treated or inhibitor-treated controls for PLY-relevant inhibitors/inhibitory conditions. Mixed glia were incubated with 2 HU/ml PLY in the presence of a vehicle or inhibitor, and the FM4-64 fluorescence change was recorded. The concentrations used are listed in the Materials and Methods and in Table 1. Abbreviations (if not self-explanatory) are C3-C3 transferase, RhoA GTPase inhibitor; NSC23766 Rac1 GTPase inhibitor; U73122-phospholipase C inhibitor; Wortmannin-PI3K inhibitor; bpV (pic)-PTEN inhibitor (last three are enzymes involved in the metabolism of PIP2 and PIP3); PP2-broad-spectrum Src-kinase inhibitor; Goe6983-broad-spectrum PKC inhibitor; GSK690693-Akt-kinase and PKC-epsilon inhibitor; ouabain-Na/K-ATPase blocker. Endocytosis by PLY is inhibited by dynamin GTPase inhibitors, potassium depletion and PI3K inhibitors; it is enhanced by PLC inhibition, Rac1 inhibition and

Na/K-ATPase block. All values represent the mean ± SEM, each dot symbol indicates an independent experiment (Dynasore experiments $n = 8$, MiTMAB experiments $n = 6$, Dyngo4a experiments $n = 8$, Amiloride experiments $n = 4$, potassium depletion experiments $n = 5$, calcium depletion experiments $n = 4$, C3 experiments $n = 4$, NSC23766 experiments $n = 4$, U73122 experiments $n = 8$, Wortmannin experiments ($n = 12$ (PLY), $n = 6$ (Wortmannin)), bpV (pic) experiments $n = 4$, bpV (pic) 12 h experiments $n = 4$, PP2 experiments $n = 6$, Goe6983 experiments $n = 4$, GSK690693 experiments $n = 4$, Ouabain experiments $n = 8$). Wilcoxon-matched pairs test (with exception of potassium depletion, NSC23766, wortmannin treatment, which were not paired experimental groups and were analyzed by Mann–Whitney $U$-test), all tests are two-tailed, exact $p$-values are indicated (if significant). Source data are provided as a Source Data file.

**Table 1 | List of the inhibitors used and rationale for their usage, signaling cascade relevance, used concentration (conc.), and 50% inhibitory concentration (IC50)**

| | Inhibitor | Type of inhibitor | Conc. | IC50 | Mechanism and relevance to possible PLY membrane effects |
|---|---|---|---|---|---|
| 1 | Dynasore | Dynamin inhibitor | 10 µM | 15 µM | Dynamin-dependent clathrin-mediated endocytosis, but also implicated in phagocytosis, flotillin-, caveolin-dependent ones. |
| 2 | MiTMAB | Dynamin inhibitor II | 10 µM | 3.1 µM | Dynamin-dependent clathrin-mediated endocytosis, but also implicated in phagocytosis, flotillin-, caveolin-dependent ones. |
| 3 | Dyngo®4a | Dynamin inhibitor | 10 µM | 5.5 µM | Dynamin-dependent clathrin-mediated endocytosis, but also implicated in phagocytosis, flotillin-, caveolin-dependent ones. |
| 4 | Amiloride | $Na^+/K^+$ exchange inhibitor | 2 mM | 2–30 µM | Macropinocytosis blocker, inhibiting the $Na^+/K^+$ exchanger, but also antagonizing the formation of submembranous acidification zones[82]. |
| 5 | NSC23766 | Rac1 GTPase inhibitor | 100 µM | 50 µM | Rac1 activity is modulated by PLY. Rac1 regulates some forms of membrane clathrin-independent endocytosis[83]. |
| 6 | C3 transferase | RhoA GTPase inhibitor | 0.1 µg/ml | 1 ng/ml | RhoA activity is modulated by PLY. RhoA can alter transferrin-mediated endocytosis[84]. |
| 7 | Ouabain | $Na^+/K^+$-ATPase & submembranous acidification blocker | 10 µM | 1 µM | Pore formation by PLY may cause cell swelling, leading to secondary activation of the $Na^+/K^+$-ATPase and to the boost of various endocytotic mechanisms[85]. |
| 8 | Go6983 | PKC inhibitor | 1 µM | 7–60 nM | Protein kinase C plays a role in some PLY effects and can boost clathrin-dependent endocytosis[86]. |
| 9 | GSK690693 | Akt/PKCε inhibitor | 10 µM | 10–20 nM | Activated Akt[87] and PKCε[88] play a role in several forms of endocytosis. |
| 10 | PP2 | Src-kinase inhibitor | 10 µM | 4–5 nM | PLY activates Src-kinases. The activated Src-kinases Fyn and Syk can stimulate endocytosis and alter cellular volume control[89]. |
| 11 | Wortmannin | PI3 kinase inhibitor | 10 µM | 2–4 nM | Membrane-associated PIP/PIP2/PIP3 signaling. Blocks the phosphorylation of PIP2 to PIP3. The latter participates in small GTPase and Akt membrane recruitment, which can alter secondarily clathrin-independent and dynamin-dependent endocytosis pathways[63]. |
| 12 | U73122 | PLC inhibitor | 10 µM | 1–5 µM | Membrane-associated PIP/PIP2/PIP3 signaling. Blocks membrane-associated PIP2 splitting into IP3 and DAG, which can activate downstream PKC and alter endocytosis[63]. |
| 13 | bpV(pic) | PTEN inhibitor | 2 µM | 31 nM | Membrane-associated PIP/PIP2/PIP3 signaling. Blocks the transition of PIP3 to PIP2 and shifts the balance of PIP2-PIP3, which affects multiple endocytosis pathways[63]. |
| 14 | $Ca^{2+}$-depletion | Ion | 0 | | Block of Ca-dependent endocytosis by some small pore-forming toxins[90]. |
| 15 | $K^+$-depletion | Ion | 0 | | Membrane hyperpolarization, clathrin-coated pit reduction[91]. |

The validation of the inhibitors is shown in Supplementary Fig. S10.

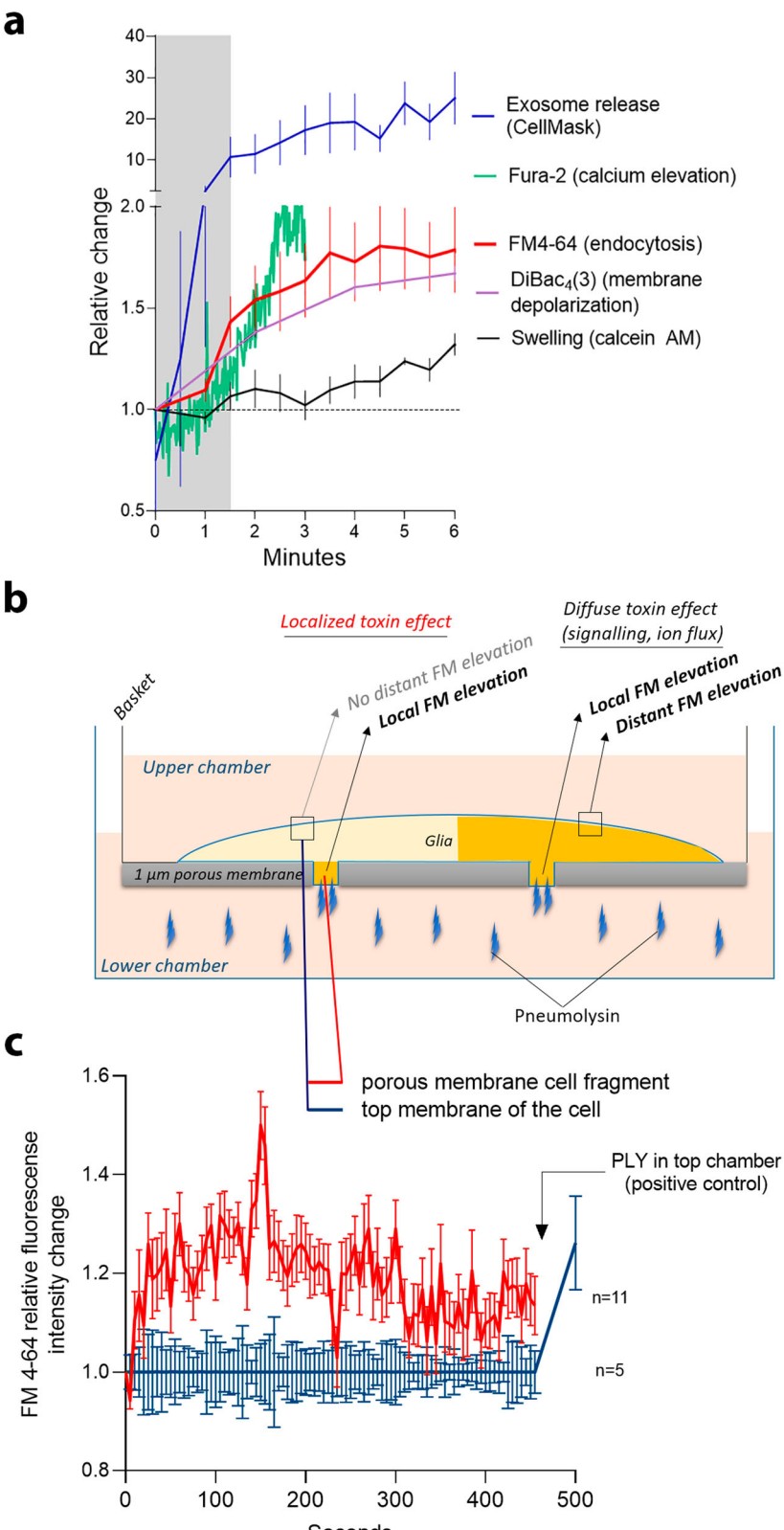

**Fig. 7 | Temporospatial features of endocytosis enhancement after challenge with recombinant PLY. a** Time frame of endocytosis change (FM4-64, normalized to timepoint 0, *n* = 6 independent experiments), exosome release (CellMask, single vesicles counted, *n* = 4 independent experiments), membrane depolarization (DiBac₄(3), fluorescence increase, normalized to timepoint 0 and to mock, *n* = 15 cells), calcium increase (Fura-2, ratiometric value, increasing with calcium increase, *n* = 4 independent experiments), and cell swelling (calcein AM, fluorescence intensity inversely normalized to timepoint 0 (the value increases when swelling increases, but in reality fluorescence falls)). **b** Scheme of the hanging basket with a 1 μm porous membrane allowing localized treatment through the pores of the underside of the cells (incubation with 2 HU/ml PLY), prestained with FM 4-64. **c** FM 4-64 fluorescence increase along the membrane with pores (red curve), normalized to the fluorescence along the top membrane (blue curve), demonstrates the localized nature of the toxin's pro-endocytotic effect. Values represent the mean ± SEM. *n* is indicated and represents single cells tested, pooled together from three independent experiments. Source data are provided as a Source Data file.

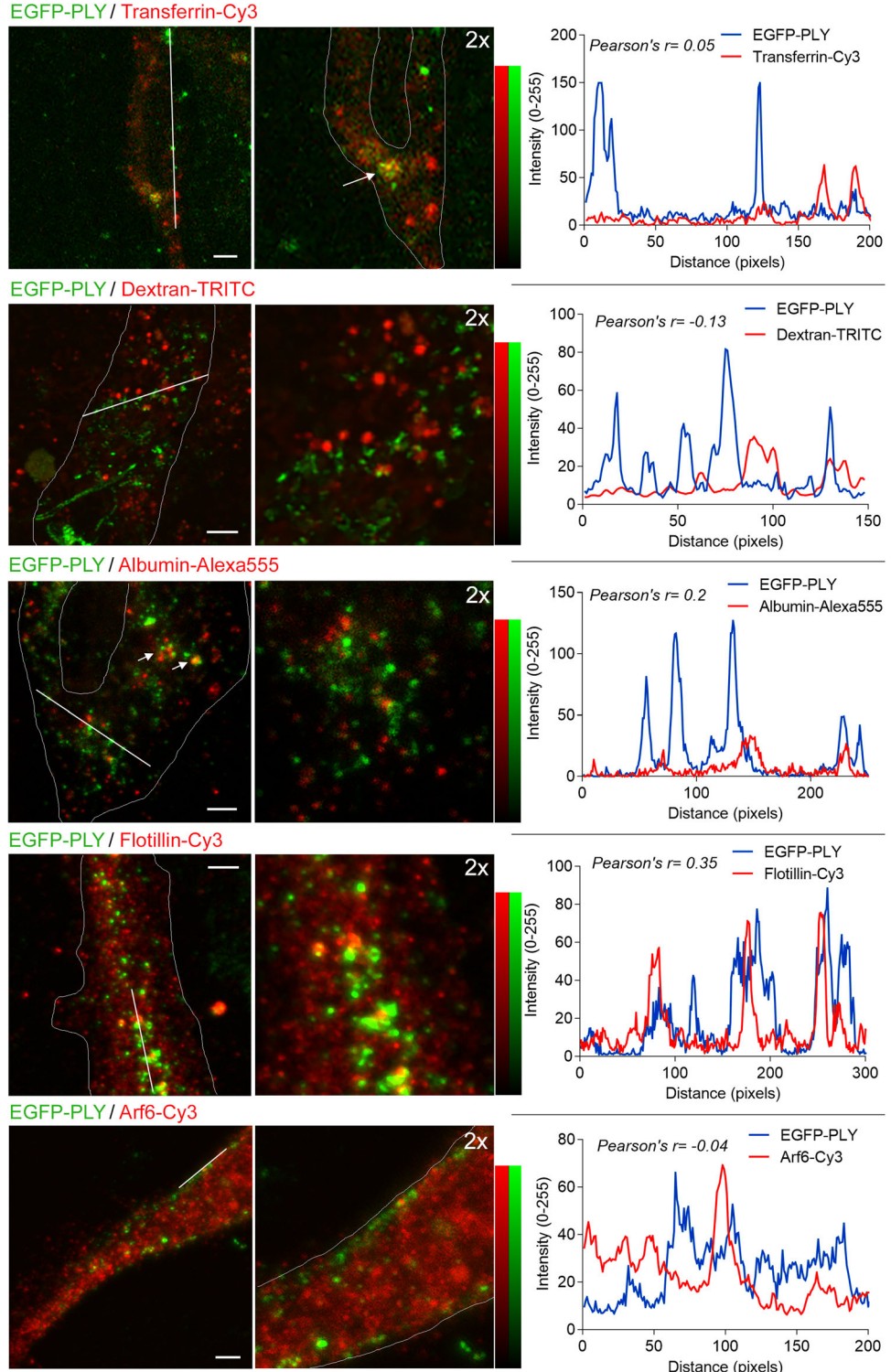

**Fig. 8 | Subcellular compartmentalization of recombinant EGFP-PLY.** Primary murine microglia were treated for 10 min with 4 HU/ml EGFP-PLY and transferrin-Cy3 (label of receptor-mediated endocytosis), dextran-TRITC (fluid-phase endosomes), or albumin-Alexa555 (caveolae) or were fixed and stained by immunocytochemistry for Arf6 (Arf-dependent endosomes) or flotillin-1 (flotillin-1-positive endosomes) (real-color images). Intensity profiles of the white line in the left column of images for both PLY and the corresponding colocalization marker are presented (altered colors for improved readability in color-vision deficiencies), and moderate colocalization (Pearson's correlation coefficient *r* > 0.3) is only observed with flotillin-1. The right column of images shows a 2-fold magnification of the area of the line profile in the corresponding left image. Scale bars: 3 μm. Images show representative cells, of experiments, repeated in triplicate with identical results.

potassium stores (the effect requires at least 2 h) and can subsequently block endocytosis by arresting coated pit formation and receptor-mediated endocytosis[62]. We cannot exclude the possibility that secondary intracellular potassium depletion contributed to the inhibition of endocytosis. All tested dynamin inhibitors diminished endocytosis. The effect of Dynasore, which has the lowest potency among all tested dynamin 1/2 GTPases, was the strongest, which suggests some additional off-target effects[41].

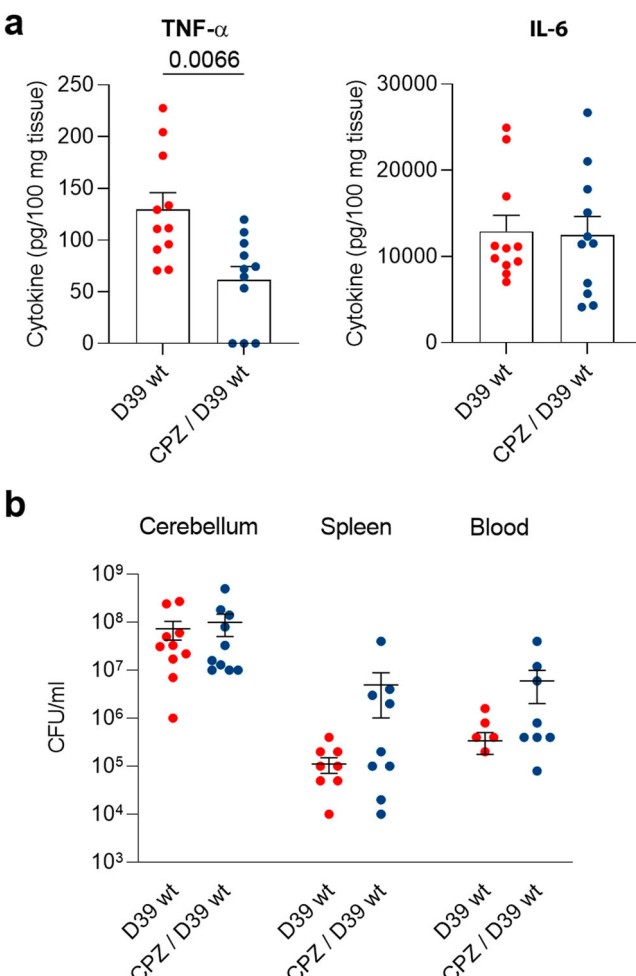

**Fig. 9 | Treatment of mice with pneumococcal meningitis with the endocytosis inhibitor chlorpromazine (CPZ). a** Reduction of TNF-α but not IL-6 release after CPZ treatment in meningitis animals (CPZ/D39 wt) versus meningitis only animals (D39 wt) 18 h after the induction of pneumococcal meningitis. **b** Bacterial CFUs in samples from cerebellum, spleen, and blood showing equivalent numbers in animals with and without CPZ treatment (zero levels in spleen and lungs are not shown due to the logarithmic scale but are included in the mean). All values represent the mean ± SEM, and each animal is indicated with a dot, $n = 11$ animals (D39 wt) and $n = 11$ animals (CPZ/D39 wt), Mann–Whitney $U$-test, all tests are two-tailed, exact $p$-values are indicated (if significant). Source data are provided as a Source Data file.

The observed inhibitory effect on PLY-enhanced endocytosis after inhibition of PI3K, which shifts the balance of PIP2 and PIP3 toward PIP2, is intriguing, but we caution in making major conclusions due to some controversies we observed. First, PLC inhibition, which should also increase the amount of PIP2, paradoxically had the opposite effect in our system: it enhanced endocytosis. The reactivation of PI3K through the turn-off of its PKC inhibition may partially explain it (for a detailed review of PIP2/PIP3 metabolism, see[63,64]). Another controversy is that the inhibition of PTEN, which dephosphorylates PIP3 to PIP2 and acts opposite to PI3K, failed to influence endocytosis (we used two treatment modalities). The metabolism of the various phosphatidylinositol phosphates is very complex, and inhibition of an enzyme may not have such straightforward effects. It is also possible that changes in PIP/PIP2/PIP3 metabolism influence the oligomerization of the toxin due to changes in the membrane lipid environment and alter endocytosis in this way.

Although we used experimentally optimized inhibitory concentrations (based on literature evidence), there are some limitations of this approach—many inhibitors may have off-target effects, one

inhibitor may have multiple targets with opposite effects, and isoforms of the targets may have different sensitivities to the inhibitors. Thus, the lack of effect of an inhibitor needs to be carefully interpreted and cannot be considered absolute proof of a cascade that is not involved.

PLY induces $Ca^{2+}$-dependent exosomal release and pore removal in glial cells[65]. The internalization processes induced by PLY, on the other hand, were $Ca^{2+}$-independent. All suggests that both processes are not mechanistically linked, although they both depend on PLY. Among the other pore-forming, cholesterol-binding cytolysins, LLO is known to enhance the internalization of *Listeria monocytogenes*[66]. LLO acts in and is released from the interior of cells. In contrast, PLY is a toxin from a bacterium that generally does not require intracellular trafficking for its life cycle or pathogenesis. PLY is normally released in the fluid phase around the cell after pneumococcal lysis. Nevertheless, some comparison with the molecular mechanism of LLO-mediated internalization enhancement shows that both mechanisms are $K^+$ dependent, although unlike LLO, PLY affects internalization in a $Ca^{2+}$-independent manner.

Apart from cholesterol, PLY has a second receptor—mannose receptor 1 (MRC1), a receptor known as a pro-endocytotic and pro-phagocytotic receptor[67]. In HEK293 cells, which do not express MRC1[68], PLY-enhanced endocytosis, which supports the role of cholesterol. PLY may use different mechanisms to enhance endocytosis in different cell types, but we believe it is not probable because the major receptor for PLY—cholesterol—is present in all cells. Domain 4 of PLY is the active interacting partner for MRC1, but it did not show any effect on endocytosis. MRC1 can also influence cytokine release, but the effect is opposite of that observed with Nod1; that is, MRC1 inhibits the proinflammatory response against D39 pneumococcus from peripheral immune cells[22].

In untreated microglia, endocytotic vesicles occurred in close proximity to membrane waves, which changed their direction shortly after toxin exposure. Such waves depend on actin dynamics, and PLY is known both as a modulator of the actin cytoskeleton and small GTPase activity[17,45]. None of the tested GTPase inhibitors blocked endocytosis by PLY, Rac1 inhibition even enhanced it, which indicates that they were not critical for the initiation of the process. While small GTPase inhibitors did not affect negatively endocytosis, the interaction between PLY and actin could still play a role. The rapid initiation of endocytosis shortly after toxin challenge, coinciding with toxin pore formation and limited to the areas of toxin exposure (not propagating outside the binding zone), suggests localized toxin/membrane effects leading to enhanced endocytosis. We suggest that this effect involves direct interaction between PLY and other membrane-associated molecules, but clarifying it falls beyond the scope of this work.

PLY enhanced dynamin-dependent inflammation-relevant endocytosis, and this was the clinically relevant finding of our work. The toxin molecule itself was internalized in a different compartment (dynamin-independent), for which we do not have evidence to be disease-relevant. We found only partial colocalization of the toxin with a specific endosome type, namely, with flotillin-1-positive endosomal vesicles, consistent with earlier works demonstrating the affinity of PLY for lipid rafts[69] (overview of the concept in Supplementary Fig. S13). The colocalization, however, suggests that flotillin can be involved in the uptake of PLY, as observed in other endocytosis processes[70]. Flotillin-1 and −2 are also involved in the internalization of two other toxins that are not pore-forming—the bacterial Shiga toxin and the plant toxin ricin[71].

The BBB remains largely impermeable to common endocytose inhibitors, limiting their clinical applicability in meningitis. The partial modulation of neuroinflammation by CPZ and the potentiation of PLY neurotoxicity at higher toxin concentrations made the drug suitable only as a proof-of-concept substance, not as a therapy. Although our findings support the role of CPZ as a modulator of endocytosis, we cannot fully exclude its possible influence on the blood–brain barrier

and the subsequent influx of serum lipids in the brain, which can also secondarily inhibit the activity of PLY[72].

The endocytosis enhancement most strongly affected proinflammatory cytokines (such as TNF-α and IL-6) and affected the tested chemokine (CXCL-2, human IL-8) to a much lesser degree. If inflammatory cytokines are more strongly affected than chemokines, this would support the deleterious neuroinflammatory mechanism based predominantly on the activation of resident innate immune cells rather than on infiltrating peripheral immune cells. Without detailed analysis of the complete cytokine and chemokine secretion profile, we remain cautious in making general conclusions.

Bacteria interact with their environment in multiple ways: by controlling their proliferative state, by masking their surface against recognition and phagocytosis, by releasing modulatory factors (such as PLY), by forming protective biofilms (where possible), and finally, by controlling the level of inflammation in the host organism. The internalization of ligands is a critical step in the activation of innate immune receptors. Exploiting intracellular trafficking as a tool for modulating inflammatory responses represents an exciting new feature of cholesterol-dependent cytolysins, providing new insights into the pathogenesis of pneumococcal infection and potentially the pathogenesis of infections by many pathogens that produce similar toxins.

## Methods

Our research complies with all state regulations, ethical commission regulation (ethical commission of the Canton Bern, see below), which approved the animal experimentation.

### Recombinant pneumolysin

Wild-type pneumolysin (PLY), N-terminally EGFP-tagged PLY (EGFP-PLY), domain 4-truncated mutant PLY (D4), Δ6 pore-deficient mutant PLY (Δ6 PLY) and W433F mutant PLY (W433F PLY)[39] (1% of the pore-forming capacity of the wild-type protein) were expressed in *Escherichia coli* BL-21 cells (Stratagene, Cambridge, UK) and purified via metal affinity chromatography. The purified toxins were tested for the presence of contaminating Gram-negative LPS using the colorimetric LAL assay (KQCL-BioWhittaker, Lonza, Basel, Switzerland). All the purified proteins showed <0.6 endotoxin units/μg of protein. The hemolytic activity was determined as described previously[37]. Briefly, one hemolytic unit (HU) was defined as the minimum amount of toxin needed to lyse 90% of 1% human erythrocytes per ml within 1 h at 37 °C. The equivalent lytic capacity in red blood cells does not explicitly correspond to the equivalent lytic capacity in other cell types[73]. For PLY, we determined a hemolytic capacity of 40,000 HU/mg. In the experiments, a concentration of 0.1 μg/ml (concentration of toxin in the CSF of meningitis patients) and 4 HU/mg were used. The plasmid encoding the nontoxic Δ6 version of the protein was constructed by site-directed mutagenesis (QuikChangeSDM Kit, Stratagene) of pET33bPLY to introduce the deletion of alanine at position 146 and arginine at position 147.

For additional LPS removal, we used a ProteoSpin Endotoxin Removal Maxi Kit (Norgen Biotek, Thorold, Canada), and LPS was determined to be below the minimal sensitivity of the LAL kits. Similarly, in some experiments, glial cells were incubated in the presence of 10 μg/ml polymyxin B (Sigma–Aldrich Chemie GmbH, Munich, Germany) to inactivate the residual amounts of LPS. Cholesterol (Sigma–Aldrich) inactivation experiments were performed in PBS at 37 °C for 15 min in a molar excess of 1:10 PLY:cholesterol, and the cholesterol stock was initially dissolved in ethanol without exceeding the final concentration of ethanol in the deactivation mix beyond 0.1%.

### Bacterial cultures and lysates

The *S. pneumoniae* bacterial strains (the wild-type D39 strain (D39 wt) and the D39 PLY-deficient mutant strain (D39 ΔPLY)) were a kind gift from Jeremy Brown from the University College London, UK. The different strains were plated on blood agar plates (Columbia agar with 5% sheep blood; Oxoid Limited, Hampshire, UK) and incubated at 37 °C under anaerobic conditions overnight. Several colonies were picked and grown to mid/late log phase ($OD_{600}$ of 0.7) in brain heart infusion broth (BHIB; Becton Dickinson and Company, le Pont de Claix, France). Then, the cultures were centrifuged and washed three times with phosphate-buffered saline (1x PBS), and serial dilutions were plated on blood agar plates according to the Miles and Misra method[74] to determine the colony-forming units (CFU) per milliliter. The bacteria were finally resuspended in PBS and subsequently diluted and applied to mixed glial cultures after incubation with 10 μg/ml ceftriaxone (Sigma–Aldrich) overnight to induce bacterial lysis. For the additional disruption of the bacterial cell walls and the release of the bacterial virulence factors, the bacterial suspensions were frozen at −80 °C for at least one night. In all experiments, the cells were treated with lysates equivalent to $1 \times 10^7$ CFU/ml. In all experiments with lysates, the mock-treated cells were treated with PBS and ceftriaxone at identical concentrations as the lysate-treated dishes.

### Cell cultures and treatments

Primary mouse astrocytes and microglia were prepared from the cortices of newborn C57BL6JRj mice (postnatal day (PD) 3–4) as mixed cultures following an established procedure[75]. Briefly, the cortices of newborn mice were dissociated into cell suspensions and plated in cell culture flasks (75 cm²) (Sarstedt AG, Nuembrecht, Germany) coated with poly-L-ornithine (PLO, Sigma–Aldrich). The growth medium, Dulbecco's modified Eagle medium (DMEM; high glutamate) (Invitrogen, ThermoFisher Scientific Inc., Waltham, MA, USA), was supplemented with 10% FCS (PAN Biotech, Aidenbach, Germany) and 1% penicillin/streptomycin (Invitrogen, ThermoFisher Scientific AG, Basel, Switzerland). Eleven to fourteen days after preparation, the cells were ready to be used in the experimental procedures. After washing the cells three times with PBS, the treatments with bacterial lysates or recombinant PLY were performed in serum-free medium to avoid PLY inactivation.

HEK293 cells (DSMZ, Braunschweig, Germany) were cultured in DMEM supplemented with 10% FCS and 1% penicillin/streptomycin.

The mixed glial cultures were treated with Dynasore (dynamin inhibitor), 10 μM; MiTMAB (dynamin inhibitor II, myristyl trimethyl ammonium bromide), 10 μM; Dyngo®4a (dynamin inhibitor), 10 μM; amiloride (inhibitor of pinocytosis and of submembranous acidification), 2 mM; ML130 (Nodinitib-1) (Nod1 inhibitor), 2 μM; GSK583 (RIP2 kinase inhibitor), 5 μM; NSC23766 (Rac1 GTPase inhibitor), 10 μM; ouabain (Na/K-ATPase blocker), 10 μM; Go6983 (broad-spectrum protein kinase C (PKC) inhibitor), 1 μM; GSK690693 (Akt and PKC epsilon inhibitor), 10 μM; PP2 (Src-kinase inhibitor (including Fyn and Syk)), 10 μM; Wortmannin (PI3 kinase inhibitor), 10 μM; U73122 (PLC inhibitor), 10 μM; bpV(pic) (PTEN inhibitor), 2 μM (all from Sigma–Aldrich); CU CPT 22 (Toll-like receptor 2 blocker), 10 μM; C34 (Toll-like receptor 4 blocker), 10 μM; and hydroxychloroquine sulfate (Toll-like receptor 9 blocker), 40 μM (all from Tocris, Bristol, UK); C3 transferase (RhoA GTPase inhibitor; Cytoskeleton Inc. Denver, CO, USA) 0.1 μg/ml; and CpG DNA, 1 μM (unmethylated DNA (ODN 1668), TLR9 agonist; TIB Molbiol, Berlin, Germany). The preincubation time was 30 min at 37 °C in serum-free medium for all inhibitors or vehicle. After pretreatment, the cells were treated with toxin or bacterial lysates in the presence of the inhibitors for the indicated time. To inhibit pneumolysin, the recombinant toxin or the bacterial lysates were preincubated with cholesterol at a concentration of 10 μg/ml (dissolved in 100% ethanol) for 30 min at 37 °C.

### Live-imaging and vital staining

Dense cultures of primary murine mixed glial cells were used to measure the increase in total endocytosis over a time course of 60–120 min. Glial cultures with monolayer densities were preincubated

in Leibovitz's L-15 $CO_2$-independent cell culture medium (Invitrogen) containing 5 μg/ml FM™ 4-64 (*N*-(3-triethylammonium propyl)−4-(6-(4-(diethylamino) phenyl) hexatrienyl) pyridinium dibromide) fluorescent dye (ThermoFisher), preloaded for 30–60 min, K-free buffer (135 mM NaCl, 2 mM $MgCl_2$, 2 mM $CaCl_2$, 5 mM HEPES (chemicals from Sigma−Aldrich)), or a control buffer with 4 mM KCl (Carl Roth GmbH + Co. KG, Karlsruhe, Germany). In all endocytosis experiments, slight variations in preincubation times, threshold settings, dye preparations, etc., can result in variations in the magnitude of enhancement of endocytosis from an experiment to an experiment. Therefore, in all experimental settings, all comparisons included only intraexperimental controls, but the normalized values differed from experiment to experiment. For the experiments with compartmentalized PLY exposure, mixed glial cells were cultured on PLO-coated membrane inserts (Thincert, Greiner Bio-One GmbH, Frickenhausen, Germany) with 1 μm pores, placed in a 24-well plate with 300 μm pores in the upper chamber and 700 μl pores in the lower chamber. Cells were stained with FM 4-64 both in the upper and lower chambers, while the toxin was present only in the lower chamber.

Membrane potential changes were tested using the membrane potential-sensitive dye $DiBAC_4(3)$ (500 nM; ThermoFisher) with excitation at 488 nm as described before[17]. Calcium influx was measured using Fura-2 AM (ThermoFisher) loaded for 30 min in primary glial cells at a concentration of 5 μM in imaging buffer, subsequently washed for 30 min, and then imaged on an IonOptix microscopy system with a 63× oil immersion objective and analyzed with its proprietary IonWizzard software version 6.3 (all from IonOptix Limited, Dublin, Ireland). The analysis of vesicle shedding was performed by staining cell membranes with CellMask Deep Red (ThermoFisher) for 20 min at 1 μg/ml, followed by high-speed z-level reconstruction on a Zeiss LSM 880 with Airyscan (Carl Zeiss AG, Oberkochen, Germany) using 63x oil immersion objectives and optical zoom between 2 and 4, resolution 512 × 512, laser illumination at 633 nm, and proprietary ZEN 2.0 software. Cell swelling was analyzed by the reduction of the fluorescence of calcein AM (ThermoFisher), prestaining the cells for 45 min at 1 μM, resolution 512 × 512, laser illumination at 488 nm.

During live-imaging, the cells were incubated in L-15 medium or imaging buffer (135 mM NaCl, 2 mM $MgCl_2$, 2 mM $CaCl_2$, 4 mM KCl, 5 mM HEPES (chemicals from Sigma−Aldrich)), and recording was carried out on an Olympus Cell^M imaging system with version 3 (build 1243) of the proprietary software (Olympus Deutschland GmbH, Hamburg, Germany) at a temperature controlled at 37 °C with a heating plate and custom-built microscope incubator with a heater and thermostat feedback loop using 10x dry objectives (if not otherwise indicated). For the analysis of endocytosis, all the images were normalized to the initial frames and timepoint 0. Furthermore, the images were secondarily normalized to the mock-treated curves, represented as a dashed line at 1. FM4-64 was visualized with a FITC filter, 3.5 to 7% intensity, 30-50 ms exposure, 1376 × 1038 pixel image size.

## Animal experiments

All animal experiments were performed in accordance with the Swiss animal experiment legislation and approved by the Animal Experiment Commission of the Canton Bern under No. BE103/2020. We used 8- to 12-week-old C57BL6JRj mice randomized into two groups containing equal numbers of male and female animals and randomized them between cages. Animals from both groups were treated and subjected to surgery in parallel (mock and treated at a time) to minimize variations. The numbers of animals in each group were determined using G-power software (see details in Statistics and reproducibility)[76]. The animals were obtained from Janvier Labs (Le Genest-Saint-Isle, France) and allowed to accommodate for 14 days in an air-conditioned animal facility with a 12/12 dark/light cycle (animals were housed in 12/12 h light/dark cycle 21 ± 2 °C, 56% relative humidity with lights turned on at 08:00 and turned off at 20:00) and provided with food and water *ad*

*libitum*, caging 4 at a time prior to the experiment without mixing male and female animals. Meningitis was induced by injecting $10^4$ CFU of D39 bacteria in 15 μl in the subarachnoid space using the following coordinates (bregma): midline, 1 mm anterior and 4 mm deep[77]. The animals were anesthetized with 5% isoflurane and maintained with 2% isoflurane (Baxter AG, Opfikon, Switzerland). Before incision, the skin was infiltrated with lidocaine/bupivacaine mix (1:1 total volume of 0.2 ml/mouse, max 4 mg/kg total; xylocaine 1%, carbostesine 0.25% obtained through the veterinary university pharmacy). Paracetamol (3.5 mg/ml, Sigma−Aldrich) was provided in the drinking water after surgery. Chlorpromazine (8 mg/kg in saline; Sigma−Aldrich) was injected intraperitoneally (200 μl) during the procedure and 9 h later. Owing to the hypothermic effect of CPZ, all animals were housed in a 37 °C temperized incubator immediately after surgery and throughout the experiment. The mice were separated into individual cages and allowed free access to food and water at the bottom of the cage. They were checked following a scoresheet at 6, 12, 15, and 18 h with the following parameters: activity, body temperature, coat, posture, grimace, body condition, and neurological status. The primary variables were the tissue cytokine levels in the left hemisphere, obtained immediately after sacrificing the animals with pentobarbital and decapitation. The tissue was weighed and homogenized in BeadBug prefilled tubes (Sigma−Aldrich) on a TissueLyser LT device (3 min, maximum speed; Qiagen AG, Hombrechtikon, Switzerland) in PBS containing cOmplete Mini EDTA-free protease inhibitor cocktail (Roche Diagnostics GmbH, Mannheim, Germany). All samples were cleared for 20 min at 15,000 × *g* and subsequently analyzed with ELISA. One cerebellar hemisphere, spleen and blood from each animal were collected for microbiological CFU count on blood agar plates using serial dilutions.

## Cytokine measurements

Murine mixed glia were plated on poly-L-ornithine-coated 24-well plates at a density of 250,000 cells/well; these populations consisted of astrocytes and microglia at approximately equal concentrations, as determined by immune staining of the mixed cultures. TNF-α, IL-6, and CXCL-2/MIP2 were selected to determine the early proinflammatory response of the mixed glia to bacterial molecular patterns. The amounts of the cytokines released were measured after 24 h of continuous incubation of the cells with live bacteria or bacterial lysates (prepared as above) at 37 °C and 5% $CO_2$ in DMEM (high glucose, GlutaMax supplement, all from Invitrogen) without phenol red and serum. We used a continuous challenge of the cells with bacteria, as this, in our opinion, mimics the situation during infection. After incubation, the supernatants were centrifuged at 5000 rpm for 5 min to remove the residual cellular and bacterial debris. The samples were stored at −20 °C until the cytokines were measured with conventional sandwich enzyme-linked immunosorbent assay against natural mouse TNF-α, IL-6, and CXCL-2/MIP2 (BioLegend ELISA MAX™, San Diego, CA, USA). The assays were carried out according to the manufacturer's instructions. The absorbance was detected at 450 nm with a standard absorbance microplate reader (EL800 and Gen5 software package, BioTek Instruments, Winooski, VT, USA). In all cytokine experiments, variations in cytokine release due to variations in cell numbers, cell culture age and bacterial lysates can be observed from an experiment to experiment. Therefore, in all experimental settings, all comparisons include only intraexperimental controls.

## Immunocytochemistry and microscopy

EGFP-tagged PLY was applied to glial cell culture for 5 min at 37 °C. Subsequently, the cells were washed once with 1x PBS and fixed with 4% formalin (Sigma) solution for 20 min. After permeabilization with 0.05% Triton X-100 (Carl Roth) in PBS on ice for 5 min, the nonspecific binding sites were blocked in 4% BSA/PBS (Carl Roth) for 30 min. The cells were incubated with the anti-flotillin-1 rabbit antibody

(1:200, ab41927, Abcam, Cambridge, UK) and anti-Arf6 rabbit antibody (1:200; ab77581, Abcam, Cambridge, UK) and with a goat anti-rabbit secondary antibody tagged with Cy3 (1:1000; 111-166-144, Jackson Jackson ImmunoResearch Europe, Cambridgeshire, UK) for 1 h at room temperature. The samples were preserved with ProLong Gold antifade reagent (ThermoFisher). For the analysis of the subcellular compartments, various fluorescent probes were used: transferrin-Cy3 for classic receptor-mediated endocytosis, dextran-TRITC for fluid-phase endocytosis and phagolysosomal labeling (Sigma–Aldrich)[78] and albumin-Alexa555 (ThermoFisher) for caveolae labeling[79]. Fluorescence images were acquired on a Zeiss LSM 880 using 63x oil immersion objectives (Carl Zeiss), resolution 1024 × 1024, PMT settings between 800 and 850 V, 488 nm laser illumination (0.5% power) for EGFP excitation and 561 nm laser illumination (5% power) for Cy3/TRITC/Alexa555 excitation.

## Transmission electron microscopy

Murine mixed glia were prepared in monolayers in PLO-coated inserts with PES membranes (Sarstedt). After treatment with 0.1 μg/ml pneumolysin (4 HU/ml) for different time points, the cell cultures were processed for electron microscopy as follows. Each insert was chemically fixed in a solution of 2.5% glutaraldehyde in 0.15 M HEPES buffer (Agar Scientific, Stansted, Essex, UK) and stored in fixative at +4 °C for several days. The total osmolarity was 700 mOsm, and the pH was 7.35. The cell cultures were then rinsed with 0.1 M sodium cacodylate buffer and postfixed for 1 h in a 1% solution of osmium tetroxide (Electron MS, Hatfield, PA, USA) in 0.1 M sodium cacodylate buffer (total osmolarity: 378 mOsm, pH 7.40). After rinsing the samples in the same buffer again, we dehydrated the samples in increasing concentrations of ethanol (70%, 80%, 96%, 100% p.a.). The samples were then left in a 1:1 mixture of ethanol:Epon 812 overnight. On the following day, the samples were embedded in Epon 812 (Fluka, Honeywell International Inc., Charlotte, NC, USA) and left to polymerize at 60 °C for 6 days.

Ultrathin sections (70 nm) were cut on a Reichert-Jung Ultracut E microtome using a diamond knife (Diatome, Biel, Switzerland). The cell layers on the membranes were cut at flat angles. The sections were placed on 200-mesh hexagonal copper grids and double stained with 1% uranyl acetate (Fluka) and 3% lead citrate (Leica Microsystems, Wetzlar, Germany).

Electron microscopy was carried out using a transmission electron microscope (Morgagni M268; FEI, Brno, Czech Republic). The cells were identified based on a set of morphological parameters, including nuclear morphology (larger, paler and oval in astrocytes and smaller, denser and with irregular shape in microglia) and cytosolic appearance (less dense, with multiple filaments, free ribosomes and fewer mitochondria; denser, cytosol richer in inclusion bodies, multiple mitochondria and multivesicular bodies)[80].

## Cytotoxicity assays

The 24 h survival of the mixed glial cultures during the treatment with the bacteria or the combined incubation with the inhibitors and bacteria was evaluated by counting all the DAPI (4′,6-Diamidino-2-phenylindol; ThermoFisher)-positive nuclei (for the total number of cells). Cells were fixed with 3% formalin solution before DAPI staining. For analysis of the acute cell permeabilization by PLY, we used a propidium iodide (PI, ThermoFisher) assay with live-imaging of permeabilized cells in the red fluorescent channel on an Olympus live-imaging system (described above). At the end of the assay, the total number of cells in each imaging field was determined by DAPI staining.

## Image analysis

Image analysis was performed using ImageJ (ver. 1.52p; NIH, Bethesda, USA) with MBF "ImageJ for Microscopy" Collection from Tony Collins and ImageJ (with Fiji add-on package, Johannes Schindelin and team). Initial fluorescence was measured in the field of interest, subtracting background fluorescence. For the analysis of endocytosis fluorescence, all the images were normalized to the initial frames and timepoint 0. Furthermore, the images were secondarily normalized to the mock-treated curves, represented as a dashed line at 1. For the nuclear cell counts, the plugin "Particle analysis/Nucleus counter" was used. For the analysis of colocalization, a line profile of the selected regions was measured in different channels and subsequently correlated using the correlation function of MS Excel (Microsoft Corporation, Redmont, USA). For movie reconstruction, stack buildup using ImageJ with subsequent conversion in a. AVI format with. JPG compression and 4 fps was performed.

## Statistics and reproducibility

Statistical analysis was performed using GraphPad Prism 9.4.1 for Windows (GraphPad Software Inc., La Jolla, CA, USA). Statistical comparison of two groups was performed by parametric (for all analyses with $n > 10$) Student's $t$-test or paired $t$-test (if performed in parallel); nonparametric (for all analyses with $n < 10$) Mann–Whitney $U$-test or paired Wilcoxon test (if performed in parallel, most of the live-imaging experiments). Statistical comparison of more than two groups was performed using one-way ANOVA followed by Dunnett's or Tukey's post-test for group comparison (according to the software recommendations)[81]. All tests were two-tailed. $p$-values below 0.05 were considered statistically significant. The results show the mean value and the standard error of the mean (SEM).

All experiments were replicated (as indicated as n) as independent experiments (raw measurements included in the repository and raw data file). All colocalization analyses were representative of at least three independent experiments, all PLY-EGFP internalization experiments (with and without dynasore) were representative of at least five independent experiments.

For all animal experiments, the sample size was determined by using G-power software (version 3.1.9.6, Franz Faul, University of Kiel, Germany), using data from a pilot experiment with the following parameters: effect size $d = 1.5$, $α$ error of probability 0.05, power 0.95. No data was excluded, the animals were randomized, maintaining the close-to-equal ratio male/female in each group. The samples were blinded before analysis and unblinded after ELISA. The investigators were not blinded to allocation during surgery and treatment.

For all cell culture experiments, all samples (controls and treatment) were collected in parallel automatically without blinding of the type of inhibitor used but blinded according to the exact position of the inhibitor and control groups during scan (using 8-multiwell chamber slides and random assignment of the groups and predefined experiment-independent scan positions). Sample groups were annotated after data collection and analysis was performed using the same evaluation macro without variations. For all in vitro experiments, each treatment was replicated four times independently and the G-power software was used to determine whether additional replicates were feasible to detect a difference trend or not without exceeding $n = 12$.

## Reporting summary

Further information on research design is available in the Nature Research Reporting Summary linked to this article.

# Data availability

All data generated or analyzed during this study are included in this published article (and its supplementary information files) and are available as datasets at https://doi.org/10.48620/63 (BORIS repository of the University of Bern). Additionally, source data are provided as a Source Data file. Confocal, fluorescent live-imaging and electron microscopy raw data are available from the corresponding author on request due to their very large size. Source data are provided with this paper.

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

## Acknowledgements

The authors would like to thank Jeremy Brown (University College London, UK) and James Paton (University of Adelaide, Australia) for providing the D39 wild-type and D39 ΔPLY strains of *S. pneumoniae*. This work was supported by the University of Bern Initiator Grant to S.H., the Swiss National Fund (SNF) Grant No. 160136 to A.I.I. and the Novartis Foundation for Biomedical Research Grant No. 20A010 to A.I.I.

## Author contributions

S.H.—designed experiments, preformed experiments, analyzed data, wrote the manuscript; C.F.—designed experiments, preformed experiments, analyzed data; F.G.—performed experiments; T.J.M.—provided materials, wrote the manuscript; A.I.I.—designed experiments, preformed experiments, analyzed data, wrote the manuscript.

## Competing interests

The authors declare no competing interests.
