## [Peer Review File · Nature Communications]

Pneumolysin boosts the neuroinflammatory response to *Streptococcus pneumoniae* through enhanced endocytosisREVIEWER COMMENTS

Reviewer #1 (Remarks to the Author):

This manuscript describes a previously unknown activity of the toxin pneumolysin. They show the toxin enhances endocytosis into glial cells and cytokine production. They suggest a role in damage of neurons during pneumococcal meningitis.

However, there appear to be some contradictory statements and results. It is stated that this PLN-induced endocytosis enhances inflammation through IL6 and TNF yet purified PLN does not enhance this in Fig 4. Similarly contradictory are the statements that the events are blocked by inhibition of dynamin in Fig 7 but not Fig 8.

The authors suggest a role for Nod1 but a pneumococcal ligand for this receptor has not been described (it is said to be Gram negative specific).

It is stated in this manuscript that literature indicates TLR2^{-/-} mice show increased inflammation. The reference cited actually shows that without TLR2, mice do not mount a defense against the bacteria at all (ie. show no appropriate inflammation) and are overwhelmed with a meaningless response only very late in disease. There is a large literature describing TLR2 ligands released by Gram positive bacteria.

While the increase in endocytosis appears to be supported by the data, the mechanism of this effect is not determined.

The discussions are a very long recap of the results without more substantive commentary.

Reviewer #2 (Remarks to the Author):

General comments:

The authors present a detailed molecular study on the neuroinflammatory response elicited by the pneumolysin of *Streptococcus pneumoniae*. They show that it is clearly dependent on endocytotic mechanisms and can be inhibited by blocking dynamin. This aspect is new and interesting and convincingly supported by the presented data. It remains however a bit unclear, why the authors focus on the two cytokines TNF- α and IL-6 in their investigation, and do not determine other proinflammatory cytokines like IL-1 β , which has also been associated with pneumolysin activity.

Minor comments:

Page numbers are missing.

Fig. 1 and Fig.2

Comparing the reaction towards the wt *S. pneumoniae* strain D39 in the F4-64 assay (Fig.1C and Fig. 2A) it is surprising, that the kinetics of the reaction are quite different. In both assays the lysate of D39 is tested. In Fig. 1C the endocytotic activity slowly increases over time and the maximum occurs after 70 min, while in Fig. 2A it is already seen after 10 min and stays at this level.

Page 6? Line 8-9

“There is evidence for the importance of PLY as a proinflammatory factor in epithelial and glial cells”

What about other CDC toxins in this regard?

Page 7? Fig. 3 A and B

The maximum values observed for TNF-alpha production after 24 hours in response to the wt D39 are quite different. In Fig. 3B it is about 1500 pg/ml with small error bars. In Fig. 3A it is 4000 pg/ml. Similar differences are observed for IL-6 production after 24 h stimulation with D39. In the assay of Fig. 3B it is 4000 pg/ml while in Fig 1A it is 2000. Please explain. Furthermore Fig. 3A and 3B show more or less the same experiment, one figure might be sufficient here.

Page 14? Endocytosis inhibition in vivo

Testing the endocytosis inhibitor CPZ in vivo in a murine model the authors could not observe the same amount of inhibition as in cell culture experiments. However the activity of pneumolysin is strongly inhibited by mouse serum (Wade et al 2014) which may complicate the interpretation of the results.

Page 18? line 1-2

“PLY is an extracellular toxin...”

This statement seems to contradict the statement on page 2 line 21 “PLY is a protein toxin released from bacteria through autolysis...”, please clarify.

Page 24 line 17

Cytokine measurements: alpha-alpha is probably a typing mistake.

REVIEWER COMMENTS

Reviewer #1 (Remarks to the Author):

This manuscript describes a previously unknown activity of the toxin pneumolysin. They show the toxin enhances endocytosis into glial cells and cytokine production. They suggest a role in damage of neurons during pneumococcal meningitis.

1. However, there appear to be some contradictory statements and results. It is stated that this PLN-induced endocytosis enhances inflammation through IL6 and TNF yet purified PLN does not enhance this in Fig 4. Similarly contradictory are the statements that the events are blocked by inhibition of dynamin in Fig 7 but not Fig 8.

Answer: Here, we believe that some misunderstanding in the figure titles is the reason for the discrepancy. Pneumolysin enhances inflammation through enhanced endocytosis only when present in bacteria, but alone, it does not have a proinflammatory effect in our system, only a proendocytotic one. Only when incubated together with lysates, where proinflammatory factors are present, is the effect revealed. We corrected the legends of the figures, as well as the text, to make clear where we applied pneumolysin in lysates and recombinant pneumolysin.

2. The authors suggest a role for Nod1 but a pneumococcal ligand for this receptor has not been described (it is said to be Gram negative specific).

Answer: Nod1 is described to bind γ -D-glutamyl-mesodiaminopimelic acid (iE-DAP) from Gram-negative bacteria and some Gram-positive bacteria (but not *S. pneumoniae*), although the original work describing this effect indicates a relatively low-affinity interaction with $K_d = 30 \mu\text{M}^1$. We were also very surprised by our findings of the role of Nod1, but the highly specific Nod1 inhibitor Nodinitib-1 is not known for off-target effects. The concept that Nod1 may play a role in pneumococcal infections is not completely new, and evidence from infant patients with Gram-positive sepsis confirms that Nod1 receptor variants may affect the disease course². Nod-like receptors sense not only peptidoglycan ligands but also small GTPase activation and ER stress³. All of this may indicate the existence of a novel Nod1 ligand or some complex cellular changes, leading to receptor activation. We amended the discussion accordingly since this is a very important point.

3. It is stated in this manuscript that literature indicates TLR2-/- mice show increased inflammation. The reference cited actually shows that without TLR2, mice do not mount a defense against the bacteria at all (ie. show no appropriate inflammation) and are overwhelmed with a meaningless response only very late in disease. There is a large literature describing TLR2 ligands released by Gram positive bacteria.

Answer: Indeed, our text was not correct. The role of TLRs in various infections and their crosstalk is very complex, and as the reviewer correctly states, we simplified the overview. TLR2 elimination impairs bacterial clearance and leads to higher inflammation in pneumococcal meningitis only later in the disease course, most likely as a compensatory response. There is ample evidence for the presence of TLR2 agonistic molecules in Gram-positive bacteria. At the same time, an interesting finding from the work of Koedel et al.⁴ suggests that in isolated peritoneal macrophages of TLR2 KO mice, no change in the inflammatory response after challenge with pneumococci is observed (contrary to the effects observed in whole animals). All of this indicates the very complex crosstalk of

innate immune receptors, and we are happy to contribute a small piece to it. We are thankful to the reviewer for this remark, and we have amended the discussion accordingly.

4. While the increase in endocytosis appears to be supported by the data, the mechanism of this effect is not determined.

Answer: This is true. In the revision, we performed multiple additional experiments, testing multiple additional modulators of endocytosis (such as Src-kinase inhibitors, modulators of the PIP2/PIP3 balance, etc.), but none showed an inhibitory effect. Next, we performed detailed live imaging, pinpointing the first 60 seconds – immediately after toxin challenge – as the moment of initial endocytosis elevation. It started shortly after the beginning of vesicle shedding and at the same time as membrane depolarization. In another setup, we incubated only portions of the microglia with toxin, showing that the enhanced endocytosis is limited to the toxin-exposed membrane fragments. The rapid onset, early change in actin membrane waves, and localized characteristics suggest a localized toxin/membrane/cytoskeleton mechanism. These additional experiments (added at the end of the document – Additional Figure 1 and Additional Figure 2, present in the text as Figure 5 and Figure 7) outline that the effect is not due to the propagation of a signaling cascade, but rather represents a localized effect at the point of toxin/membrane interaction. We are currently investigating the exact molecular mechanisms, which include interactions between the toxin and the complex including actin, dynamin and BAR-containing proteins. Unfortunately, this information, which is still in process of analysis, is extensive and would not fit the format of the current manuscript. With our new experiments exploring the compartmentalization of the response, however, as well as additional tests of multiple endocytosis-relevant signaling cascades, we believe we provide the missing mechanistic interpretation for this manuscript.

5. The discussions a very long recap of the results without more substantive commentary.

Answer: A valid remark. We completely reworked the discussion and eliminated any unnecessary results data, complementing it with our mechanistic data interpretation.

Reviewer #2 (Remarks to the Author):

General comments:

1. The authors present a detailed molecular study on the neuroinflammatory response elicited by the pneumolysin of *Streptococcus pneumoniae*. They show that it is clearly dependent on endocytotic mechanisms and can be inhibited by blocking dynamin. This aspect is new and interesting and convincingly supported by the presented data. It remains however a bit unclear, why the authors focus on the two cytokines TNF-alpha and IL-6 in their investigation, and do not determine other proinflammatory cytokines like IL1-β, which has also been associated with pneumolysin activity.

Answer: We focused on TNF-alpha and IL-6 only because these two cytokines correlate best with the clinical severity and prognosis of pneumococcal meningitis and are most likely the two key pathogenic cytokines in this condition⁵. Completely mature IL1-β release from microglial cells in culture requires an additional purinergic receptor activation step, which makes analysis methodologically more variable and complex than that of the other inflammatory cytokines. The remark of the reviewer, however, pinpointed another gap – we did not analyze the effect of endocytosis and Nod1 on chemokine stimulation. Therefore, we performed additional experiments

to test the release of the major chemokine CXCL-2/MIP2. It was also endocytosis- and Nod1-dependent, although the effect was not as prominent as with TNF-alpha and IL-6. The experimental evidence for CXCL-2 fit exactly our conclusion regarding the other cytokines.

Minor comments:

1. Page numbers are missing.

Answer: Corrected.

2. Fig. 1 and Fig.2

Comparing the reaction towards the wt *S. pneumoniae* strain D39 in the F4-64 assay (Fig.1C and Fig. 2A) it is surprising, that the kinetics of the reaction are quite different. In both assays the lysate of D39 is tested. In Fig. 1C the endocytotic activity slowly increases over time and the maximum occurs after 70 min, while in Fig. 2A it is already seen after 10 min and stays at this level.

Answer: We used different lysate preparations in these different sets of experiments. Although all treatment and control groups were performed in tandem for each experiment, in different sets, we used different lysate preparations. We determined that in some of them (in Fig. 2), stronger autolysis probably occurred before the subsequent controlled lysis of the rest of the bacteria, elevating the amount of the released pneumolysin. To eliminate these variations, we prepared new lysates for revision and performed the endocytosis tests in Fig. 1, Fig. 2, and Supplementary Fig. S1 again in quadruplicate or more. The results fully reproduced the outcome from the original experiments, but the variations in the kinetics were eliminated.

3. Page 6? Line 8-9

“There is evidence for the importance of PLY as a proinflammatory factor in epithelial and glial cells”

What about other CDC toxins in this regard?

Answer: Pneumolysin is the best-studied proinflammatory CDC, but evidence exists for the proinflammatory effects of others, such as perfringolysin and listeriolysin^{6,7}. We added the corresponding references to the revised version.

Page 7? Fig. 3 A and B

The maximum values observed for TNF-alpha production after 24 hours in response to the wt D39 are quite different. In Fig. 3B it is about 1500 pg/ ml with small error bars. In Fig. 3A it is 4000 pg/ml. Similar differences are observed for IL-6 production after 24 h stimulation with D39. In the assay of Fig. 3B it is 4000 pg/ml while in Fig 1A it is 2000. Please explain.

Answer: It is known that primary cultures, especially mixed glial cultures, are heterogeneous regarding the number of microglia and the ratio of microglia/astrocytes, despite standard preparation. Sometimes small differences, such as harvesting 3, 4, or 5 days after birth, may affect the exact ratio. Additionally, despite very careful titration of the number of bacteria in the lysate, there are variations from preparation to preparation. This is the reason why values differed from one experiment to another. We want to emphasize, however, that each experimental replicate was performed with the same culture and the same bacterial preparation for all experimental groups within the experiment, without exception. In this way, in each repeat, the internal control was

present. Statistically, only paired tests were performed. This additional information was also added to the statistics section.

Furthermore Fig. 3A and 3B show more or less the same experiment, one figure might be sufficient here.

Answer: Indeed, completely justified remark. Corrected.

Page 14? Endocytosis inhibition in vivo

Testing the endocytosis inhibitor CPZ in vivo in a murine model the authors could not observe the same amount of inhibition as in cell culture experiments. However the activity of pneumolysin is strongly inhibited by mouse serum (Wade et al 2014) which may complicate the interpretation of the results.

Answer: Differences in the potency of inhibition are expected, especially when the blood-brain barrier permeable inhibitor CPZ is not as optimal as the in vitro inhibitors of endocytosis. Indeed, serum cholesterol inhibits the effect of pneumolysin, but in the brain, where we tested cytokine levels, serum was not present, or at least not in the early phases of the disease when the blood-brain barrier was still intact. Nevertheless, we cannot completely exclude such a mechanism (modulation of the BBB), a point which we added to the discussion.

Page 18? line 1-2

“PLY is an extracellular toxin...

This statement seems to contradict the statement on page 2 line 21 “PLY is a protein toxin released from bacteria through autolysis...”, please clarify.

Answer: It is our mistake; we meant that the toxin acts normally from outside the cells of the host organism. We did not specify this explicitly, which is, indeed, misleading – in bacteria, it is intracellular. We have corrected this accordingly.

Page 24 line 17

Cytokine measurements: alpha-alpha is probably a typing mistake.

Answer: Corrected.

New references:

1. Laroui, H. *et al.* l-Ala- γ -d-Glu-meso-diaminopimelic Acid (DAP) Interacts Directly with Leucine-rich Region Domain of Nucleotide-binding Oligomerization Domain 1, Increasing Phosphorylation Activity of Receptor-interacting Serine/Threonine-protein Kinase 2 and Its Interaction with Nucleotide-binding Oligomerization Domain 1. *J. Biol. Chem.* **286**, 31003 (2011).
2. Sampath, V. *et al.* A Potential Role for the NOD1 Variant (rs6958571) in Gram-Positive Blood Stream Infection in ELBW Infants. *Neonatology* **112**, 354–358 (2017).
3. Wiese, K. M., Coates, B. M. & Ridge, K. M. The Role of Nucleotide-Binding Oligomerization Domain-Like Receptors in Pulmonary Infection. *Am. J. Respir. Cell Mol. Biol.* **57**, 151–161 (2017).
4. Koedel, U. *et al.* Toll-like receptor 2 participates in mediation of immune response in experimental pneumococcal meningitis. *J. Immunol.* **170**, 438–44 (2003).

5. Albrecht, L.-J. *et al.* Lack of Proinflammatory Cytokine Interleukin-6 or Tumor Necrosis Factor Receptor-1 Results in a Failure of the Innate Immune Response after Bacterial Meningitis. *Mediators Inflamm.* **2016**, 1–12 (2016).
6. Dewamitta, S. R. *et al.* Listeriolysin O-dependent bacterial entry into the cytoplasm is required for calpain activation and interleukin-1 α secretion in macrophages infected with *Listeria monocytogenes*. *Infect. Immun.* **78**, 1884–1894 (2010).
7. Suzaki, A. *et al.* Pathogenic Characterization of *Clostridium perfringens* Strains Isolated From Patients With Massive Intravascular Hemolysis. *Front. Microbiol.* **12**, (2021).

New figures.

Additional Figure 1.

Time-lapse imaging of microglial membrane alterations after recombinant PLY exposure. A. High-resolution video microscopy of microglia in transmission and fluorescent imaging (FM 4-64). The fragment in the first image is magnified in the following series (0 s – time-point of treatment). (Additionally, see Supplementary movie M2). Before treatment, the membrane demonstrates regular balanced wave dynamics. Following challenge with 2 HU/ml PLY, the wave dynamics redirect toward the body of the cell and slow down, followed by retraction and clustering of the vesicular structures in

the center of the cell. White arrows indicate newly occurring endosomal vesicles, and the red arrow indicates released membrane vesicles. All newly occurring FM-positive endosomal structures are observed adjacent to membrane waves. Scale bar: 10 μm . B. Electron microscopy of the microglial surface demonstrates a massive increase in membrane protrusions, with a strong increase in the number of vesicular structures (red arrows). Scale bar: 500 nm.

Additional Figure 2.

A. Timeframe of endocytosis, exosome release, membrane depolarization, calcium increase, and cell swelling. B. Scheme of the hanging basket with a 1 μm porous membrane allowing localized treatment through the pores of the underside of the cells (incubation with 2 HU/ml PLY), prestained with FM 4-64. C. FM 4-64 fluorescence increase along the membrane with pores (red curve), normalized to the fluorescence along the top membrane (green curve), demonstrates the localized nature of the toxin's pro-endocytotic effect. Values represent the mean \pm SEM.

REVIEWER COMMENTS

Reviewer #1 (Remarks to the Author):

This manuscript focuses on what the authors indicate is a new mechanism of endocytosis of the pneumococcal cytotoxin PLY and the subsequent response of a mixture of brain cells. The authors are clearly very capable investigators of uptake mechanisms using imaging. However, the writing style is very convoluted and many inaccuracies and inconsistencies in the presentation prevent a valid assessment at this stage of what cells are involved, what other paths were excluded and what the role of this new path is compared to classical inflammation.

Glia are phagocytic. There is no distinction made to say that this is not phagocytosis.

There is no description of what procedure was used to separate various cells in the suspensions of dissociated mouse cortices into "glial cells". How were vascular and neuronal cells removed? Were microglia retained? Methods say the cells used are astrocytes yet key cells in inflammation of CNS are microglia. Then in Fig 3 it seems that there are both astrocytes and microglia. Then in later Figures, some are done with astrocytes and some microglia. How these were separated into distinct cultures is again not clear. These inconsistencies make conclusions difficult.

Bacterial lysates have a myriad of components that cause glia to activate and produce cytokines. This is not new. In a 24 hr incubation, the array of cytokines will be much broader than the 4 focused on in Fig 1. IL-6 and TNF production begins much sooner than 24 hrs and thus it is not clear if this data is relevant to higher levels which would appear earlier. It is not clear if this focus on a subset of cytokines and an unusual time point skew the results.

Fig S4. The fact that inhibition of TLRs not only did not decrease the production of cytokines by bacterial lysates but significantly increased it is very suspect. One of the most potent bacterial components is the cell wall released during the treatment of bacteria with antibiotics (as done in this case). There must be some aspect of the assay used in this manuscript that eliminates that response (? 24 hrs is too late; preparation of the lysate removes cell wall?).

Fig S3 has an incorrect figure legend

Fig S5: The legend doesn't indicate what the white and gray bars represent. The legend states that removing LPS from PLY prep removes all cytokine stimulation. Yet PLY is known to stimulate cytokine production. How is this explained?

Table 1 is critical to the argument that inhibition of all other mechanisms of uptake did not affect their type of endocytosis and therefore the whole process is novel. However, the display of the Table shows only a snap shot of one corner of the table. Thus, the validity of the claim can't be assessed.

To verify that inhibitors failed to block a response (and thus support that main conclusion that they have discovered a new mechanism), a positive and negative control should be presented for each inhibitor. While this may not need to be in the primary figures, it should at least be in supplementary material.

Reviewer #2 (Remarks to the Author):

In my view the authors have adequately addressed all the comments and concerns that were raised.

Reviewer #1 (Remarks to the Author):

1. This manuscript focuses on what the authors indicate is a new mechanism of endocytosis of the pneumococcal cytolysin PLY and the subsequent response of a mixture of brain cells. The authors are clearly very capable investigators of uptake mechanisms using imaging. However, the writing style is very convoluted and many inaccuracies and inconsistencies in the presentation prevent a valid assessment at this stage of what cells are involved, what other paths were excluded and what the role of this new path is compared to classical inflammation.

We agree that the story and content is complex and requires improved wording. We worked to improve it and simplify the presentation. Following the remarks of the reviewer, we realized a major issue – the details of the mixed glial cultures, which are well-known to the researchers in the field, may look confusing for people outside the field. We added necessary references, we outlined the benefits of isolating native astrocytes/microglia together as mixed cultures and exact methodology to identify microglia in live imaging experiments. To improve the logic of the manuscript, we extended the summary table and added an overview of the endocytosis cascades and mechanisms, which were excluded, explaining why we chose exactly these inhibitors and pathways. All these endocytosis-relevant cascades have their relevance to PLY-mediated effects.

2. Glia are phagocytic. There is no distinction made to say that this is not phagocytosis.

The fluorescent intensity of the FM dyes generally does not change substantially in phagosomes, additionally our electron microscopy analysis detected the elevation of vesicles with a size below 500 nm, as most of them were under 100 nm in size, that means well below the 500 nm threshold for phagosomes. This was one of the main reasons to go for an electron microscopy too. Finally, the effect was observed in non-phagocytotic cells as well (e.g., astrocytes), which are also known to be proinflammatory in the brain. Now we add a clarification of this question.

3. There is no description of what procedure was used to separate various cells in the suspensions of dissociated mouse cortices into “glial cells”. How were vascular and neuronal cells removed? Were microglia retained? Methods say the cells used are astrocytes yet key cells in inflammation of CNS are microglia. Then in Fig 3 it seems that there are both astrocytes and microglia. Then in later Figures, some are done with astrocytes and some microglia. How these were separated into distinct cultures is again not clear. These inconsistencies make conclusions difficult.

Working years on glial physiology and pathology, it is our mistake that for people outside the field it can be sometimes confusing why we use mixed glia. Isolation of primary glial cells (astrocytes and microglia) has been established for decades in the form of mixed glial cultures and is widely used according to the standard protocol, which we also implement. Both cell types are best isolated together, in a physiological ratio, mimicking reliably the non-neuronal environment. Additionally, both cell types are functionally tightly coupled. The brain tissue is used for simultaneous isolation of both cell types and it is nearly impossible to separate both cell types at the stage [1]. Their ratio is physiological, the interaction between astrocytes and microglia remains of critical importance for modeling best the inflammatory milieu of the brain. Further, the conditions in the co-culture inhibit the growth of other cell types such as neurons. Technically, astrocytes form a monolayer at day 10-14, while microglial cells move on the top of them, making their morphological discrimination during imaging easy. Exactly such imaging identification was performed to identify the microglia and the astrocytes. We now include very detailed protocol description of this imaging separation, including methodological references. We believe this will be of importance for people outside the immediate neuroscience field.

Bacterial lysates have a myriad of components that cause glia to activate and produce cytokines. This is not new. In a 24 hr incubation, the array of cytokines will be much broader than the 4 focused on in Fig 1. IL-6 and TNF production begins much sooner than 24 hrs and thus it is not clear if this data is relevant to higher levels which would appear earlier. It is not clear if this focus on a subset of cytokines and an unusual time point skew the results.

To answer this question, we added additional information showing the timeframe of TNF- α and IL-6 temporal dynamics of release upon challenge with bacterial lysates as a supplementary figure S6. The data clearly shows that the synthesis indeed starts earlier, but it maintains its increase until 24 h in a linear manner. For PLY-deficient strains, indeed, some earlier plateau effects at 12 h have been already observed, without diminishing the relevance of the findings. Thus, the timepoint of 24 h seems suitable and optimal without the risk of skewing the results. We focused on TNF- α and IL-6 as a major readout, because clinical evidence shows that these two cytokines correlate best with the prognosis and the clinical course of bacterial meningitis [2].

Fig S4. The fact that inhibition of TLRs not only did not decrease the production of cytokines by bacterial lysates but significantly increased it is very suspect. One of the most potent bacterial components is the cell wall released during the treatment of bacteria with antibiotics (as done in this case). There must be some aspect of the assay used in this manuscript that eliminates that response (? 24 hrs is too late; preparation of the lysate removes cell wall?).

The data we present, although surprising and even contradictory at first look, fits the observations of the pneumococcal infection in TLR2 ko animals, for example, where the lack of the receptor leads to stronger proinflammatory response despite the opposite cell culture data [3]. Until now, there is no satisfactory explanation why in such animal experiments, where whole pathogens interact with the immune system, the lack of a specific TLR does not correlate with a weaker inflammatory response, but even enhances it. One of the explanations may be the existence of additional factors, such as PLY, which can alter the intracellular trafficking as we describe in this manuscript. We present accurately experimental data, obtained from precisely designed and controlled experiments, and we avoid to overinterpret it outside the specific question we focus on. We do not observe any excessive toxicity or other culture changes, actually we see proper release of inflammatory factors, which indicates good cell status, therefore we are confident our experiments are well-conducted.

Fig S3 has an incorrect figure legend

Corrected.

Fig S5: The legend doesn't indicate what the white and gray bars represent. The legend states that removing LPS from PLY prep removes all cytokine stimulation. Yet PLY is known to stimulate cytokine production. How is this explained?

We are grateful for the remark, and we corrected the legend accordingly. Indeed, our finding here is important to our inflammation data interpretation, because apparently minute amounts of LPS contamination may have a profound effect on the data in co-stimulation experiments. We are aware that this raises many questions about the overinterpretation of published data by many scientists with description of proinflammatory effects of PLY. At the same time, such cautious remark is essential for correct experimental setups on inflammation with PLY, where such complementary effects may have a role. We believe that this critical remark will make many researchers aware of some traces in their preparations that may have surprising effects and "contaminate" otherwise correctly obtained data.

Table 1 is critical to the argument that inhibition of all other mechanisms of uptake did not affect their type of endocytosis and therefore the whole process is novel. However, the display of the Table shows only a snap shot of one corner of the table. Thus, the validity of the claim can't be assessed.

We think there was a kind of a bug with the table conversion (only a fragment of it appeared in the draft), now we include a very detailed table with references and exact argumentation why we use specific inhibitors, including a summary of the validation data from the supplementary figure S11. In this way, the same information can be obtained from several sources in the manuscript.

To verify that inhibitors failed to block a response (and thus support that main conclusion that they have discovered a new mechanism), a positive and negative control should be presented for each inhibitor. While this may not need to be in the primary figures, it should at least be in supplementary material.

We focused specifically on the effects of the inhibitors in the revised version, carefully addressing the concerns of the reviewer. All “data not shown” are now presented as graphs in a completely revised figure 6 (normalized curves), raw non-normalized data (supplementary figure S7) and repository data file. In the previous version, we have underestimated that some of the tested inhibitors that did not inhibit PLY-induced endocytosis enhanced it - the Rac1 inhibitor, the PLC inhibitor, and the inhibitor of the Na/K ATPase. This has mechanistic importance too and is evidence of the biologic activity of the inhibitors. Therefore, it is now discussed too. For one of the inhibitors, Wortmannin (a PI3K inhibitor), we performed additional experiments with a higher concentration (10 μ M instead of 200 nM) following the recommendation of an expert in the field of PIP/PIP2/PIP3 metabolism and we observed an inhibitory effect on PLY-enhanced endocytosis, which we believe contribute to the mechanistic interpretation of the data. All this is now added in the figure and text data and interpreted in the discussion. In the supplementary figure S11, we validate the activity of each of the inhibitors we used – some inhibitors (e.g., Dynasore, Dyngo4a, bpV(pic), GSK690693) influence the background fluorescence in the control treatments; other inhibitors influence the PLY-enhanced endocytosis (MITMAB, amiloride, U73122, Wortmannin, ouabain and NSC23766). Only four inhibitors needed additional verification of their activity (positive control). All of them are known to influence the motility of macrophages/microglia, which we analyzed in our transmission images from the live imaging experiments – all of them (amiloride, PP2, Go6983 and C3 transferase) inhibited motility as described in the literature. The inhibitor validation figure S11 is specifically designed to demonstrate this validation. We were very careful to not overstate our endocytosis findings and tune down the tone, to outline controversies (for example the lack of effect of the PTEN blocker, which should have acted just opposite to PI3K). Technically, now all curves are color-coded instead of shape-coded, which improves the readability.

1. Chen SH, Oyarzabal EA, Hong JS. Preparation of Rodent Primary Cultures for Neuron–Glia, Mixed Glia, Enriched Microglia, and Reconstituted Cultures with Microglia. *Methods Mol Biol.* NIH Public Access; 2013;1041:231.

2. Laborada G, Rego M, Jain A, Guliano M, Stavola J, Ballabh P, et al. Diagnostic Value of Cytokines and C-reactive Protein in the First 24 Hours of Neonatal Sepsis. *Am J Perinatol.* 2003;20:491–502.

3. Letiembre M, Echchannaoui H, Ferracin F, Rivest S, Landmann R. Toll-like receptor-2 deficiency is associated with enhanced brain TNF gene expression during pneumococcal meningitis. *J Neuroimmunol.* *J Neuroimmunol;* 2005;168:21–33.

REVIEWERS' COMMENTS

Reviewer #1 (Remarks to the Author):

This revision is substantially clearer. The added experiments present a thorough analysis that supports the conclusions.

What remains is a confusion of what the authors are trying to conclude. Examination of the abstract demonstrates this problem. They make 3 statements:

1. "Pneumococcal lysates enhanced dynamin-dependent endocytosis, and dynamin inhibition blocked the neuroinflammatory response confirming the importance of ligand internalization ... Knocking out pneumolysin eliminated their pro-endocytic effects and diminished the neuroinflammatory response."

2. "Pneumolysin-GFP was rapidly but dynamin-independently internalized."

3. "Chlorpromazine, an endocytosis inhibitor that crosses the blood-brain barrier,... demonstrated a diminished neuroinflammatory response"

1 and 2 seem to say that PLN has two activities. It doesn't say what the second activity leads to. ? inflammation? 3 doesn't say whether chlorpromazine inhibits 1 or 2.

The entire manuscript keeps flipping back and forth and, even for a reviewer who considers themselves conversant with the field, it is still confusing. The goal would be to clearly describe 2 activities of PLN and fully describe how the each contribute to disease

What remains is a confusion of what the authors are trying to conclude. Examination of the abstract demonstrates this problem. They make 3 statements:

1. "Pneumococcal lysates enhanced dynamin-dependent endocytosis, and dynamin inhibition blocked the neuroinflammatory response confirming the importance of ligand internalization ... Knocking out pneumolysin eliminated their pro-endocytic effects and diminished the neuroinflammatory response."

2. "Pneumolysin-GFP was rapidly but dynamin-independently internalized." 3. "Chlorpromazine, an endocytosis inhibitor that crosses the blood-brain barrier,... demonstrated a diminished neuroinflammatory response"

1 and 2 seem to say that PLN has two activities. It doesn't say what the second activity leads to. ? inflammation? 3 doesn't say whether chlorpromazine inhibits 1 or 2.

The entire manuscript keeps flipping back and forth and, even for a reviewer who considers themselves conversant with the field, it is still confusing. The goal would be to clearly describe 2 activities of PLN and fully describe how the each contribute to disease

Indeed, the statements we make in the abstract and throughout the manuscript about the internalization of the toxin itself is confusing and, frankly, unnecessary. The very mechanism of toxin internalization, although interesting from a cell biological point of view, carries no additional information to the pathology of the disease. The important message is the enhancement of dynamin-dependent endocytosis by pneumolysin and its importance as a driver of fatal neuroinflammation. The effect of chlorpromazine is clearly on clathrin-mediated endocytosis (references in the literature), which is dynamin dependent.

Now, we removed these uncertainties from the abstract and from the body of the manuscript, and we reworded parts of the results and the discussion section to keep any confusion at bay.